# STAT: Skill-Targeted Adaptive Training

**Yinghui He**[*] **Abhishek Panigrahi**[*] **Yong Lin** **Sanjeev Arora**
Princeton Language and Intelligence, Princeton University
{yh0068, ap34, yl7690, arora}@princeton.edu

## Abstract

Language models often show little to no improvement (i.e., "saturation") when trained via vanilla supervised fine-tuning (SFT) on data similar to what they saw in their training set (e.g., MATH). We introduce a new fine-tuning strategy, STAT, to train such a student model by using the metacognition ability of a stronger large language model (LLM) as the teacher. The teacher uses the task dataset to create a list of skills needed for the task, and then labels each data point with its required skills (Didolkar et al., 2024). By monitoring the student's answers, the teacher creates a `Missing-Skill-Profile` for the student, tracking how often they failed to apply each skill in their responses. We use this idea to build a modified training set in one of two ways. In *STAT-Sel*, the teacher uses an existing set of training examples but adaptively reweights them according to the `Missing-Skill-Profile`. In *STAT-Syn*, the teacher synthesizes additional examples involving missing skills. Across extensive experiments on Llama and Qwen models, our methods yield improvements of up to $7.5\%$ on MATH, whereas SFT provides only limited gains. Furthermore, STAT enhances performance on out-of-distribution benchmarks (e.g., AIME24/25, AMC23, etc.) by an average of $4.6\%$. Crucially, we find that STAT is complementary to RL via GRPO (Shao et al., 2024): after the model is improved using STAT to address skill gaps, GRPO continues to add further gains. We conclude that skill-targeted adaptive training should broadly improve current training pipelines. [1]

| Models | MATH | MATH$^D$ | MATH$^2$ | GSM8K | AMC23 | MATH-perturb | | AIME | | Avg. |
|---|---|---|---|---|---|---|---|---|---|---|
| | | | | | | simple | hard | 2024 | 2025 | |
| Llama-3.2-3B-Instruct | 44.0 | 18.2 | 21.9 | 73.0 | 21.7 | 33.7 | 12.2 | 33.3 | 16.7 | 30.5 |
| +SFT | 44.8 | 22.9 | 21.0 | 75.1 | 20.8 | 33.0 | 12.2 | 30.0 | 20.0 | 31.1 |
| +GRPO | 45.4 | 24.4 | 23.3 | 77.4 | 25.8 | 38.4 | 11.8 | 33.3 | 6.7 | 31.8 |
| **STAT-Sel** | **51.5** | 26.6 | 25.7 | **80.2** | **24.7** | **39.8** | 13.3 | **43.3** | 23.3 | **36.5** |
| **STAT-Syn** | 50.2 | **31.7** | **26.2** | 79.2 | 23.9 | 39.1 | **14.7** | 40.0 | **30.0** | **37.2** |

Table 1: STAT significantly enhances the performance of Llama-3.2-3B-Instruct on various math benchmarks by targeting its missing skills in solving MATH. See Table 3 for full evaluation results.

## 1 Introduction

Language models have demonstrated remarkable success at acquiring knowledge from large-scale natural text corpora through the next-token prediction objective (Shannon, 1951). Subsequent supervised fine-tuning on curated data using the same objective then leads to strong performance on domain-specific tasks such as mathematics. However, this process is often inefficient and data hungry (Kaplan et al., 2020; Muennighoff et al., 2023; Zhang et al., 2024a; Villalobos et al., 2024), with models quickly reaching a *saturation point* for a fixed dataset whereby further training does not help performance. Several works have suggested that this saturation happens because the loss is an average over data points, causing the training signal to diminish as the model becomes adept at most of the training examples (Chen et al., 2023; Xie et al., 2023a; Lin et al., 2024; Tong et al., 2024;

---

[*]Equal contribution.
[1] Code: https://github.com/princeton-pli/STAT.

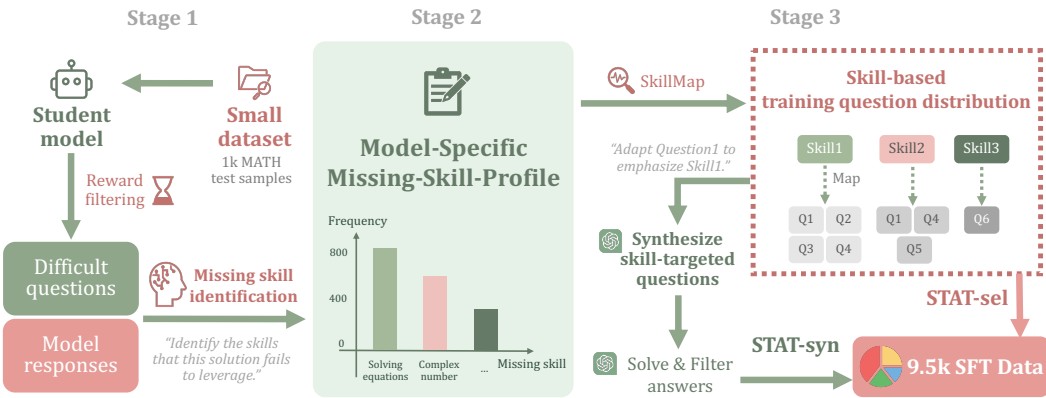

Figure 1: STAT is a three-stage skill-based data selection/generation method for supervised fine-tuning (SFT). **Stage 1:** Identify difficult questions for each model using reward filtering on model responses. **Stage 2:** Use frontier LLMs to analyze the model responses and build a model-specific `Missing-Skill-Profile`. **Stage 3:** Use a pre-constructed `Skill-Map` to map the missing skill distribution to a training question distribution, which constitutes the STAT-Sel data. STAT-Syn synthesizes new questions using frontier LLMs targeted to the missing skills.

Jiang et al., 2024; Xue et al., 2025; Zhang et al., 2025). In addition, there is a mismatch between the "average" next-token prediction loss used during training and the auto-regressive generation process used to evaluate performance. As a result, the average loss may fail to capture the specific generation errors that remain in a saturated model (Arora et al., 2022; Fang et al., 2024).

To tackle this saturation, prior works have shown that adapting the training data distribution can boost performance on inference-time tasks. The key idea is to focus the next-token prediction loss on an adapted set of examples targeted towards good generation (Xia et al., 2024; Yu et al., 2024b; Lin et al., 2024). This is primarily done by using embeddings or gradient-based estimates to pick training examples most relevant to reducing loss on a reference validation set. While these methods show benefits, they remain anchored to validation-set loss, which is only a coarse proxy for a model's actual generation errors. In fact, our experiments reveal that embedding-based methods, which adapt training data by measuring similarity to validation questions the model fails on, can be ineffective (Section 3.2) on saturated models that have undergone extensive post-training, e.g., Llama-instruct models.

We propose to address the saturation problem by drawing inspiration from pedagogical practices rooted in cognitive science, which customize training to specially target the student's skill-deficiencies (Bandura & Walters, 1977; Hattie & Timperley, 2007).

> *How can we effectively use today's strong teacher models to design better training strategies to help small models overcome their saturation?*

We turn to a growing line of research in LLM meta-cognition (Didolkar et al., 2024; Kaur et al., 2024), which leverages the predictive abilities of frontier LLMs to reason about the high-level skills required to solve a given task, as well as the skills actually being used in the student's answer. Thus, in principle, frontier LLMs can act as the teacher who guides the training process of the student model, actively monitoring the student's competence on individual skills and adjusting their training examples.

**Informal description of data design:** Our pipeline starts with a list of relevant skills for the problem created via teacher metacognition (Didolkar et al., 2024), and adds three stages. First, we use the teacher to evaluate the student model on a small validation set of questions and use a reward model to identify the questions that are difficult for the student. Second, we create a `Missing-Skill-Profile` by using the teacher to check the missing skills in the model responses. Our first method **STAT-Sel** simply up-weights training examples using the `Missing-Skill-Profile`; in effect, this guides the student to focus on their deficiencies. Our second (more expensive) method **STAT-Syn** uses the teacher to generate synthetic training

data using in-context examples from the validation set associated with a list of deficient skills in `Missing-Skill-Profile`.

**Key findings:** Applying STAT-Sel and STAT-Syn teaching on Llama and Qwen models with the popular MATH (Hendrycks et al., 2021) data shows the following:

1. **Substantial in-distribution gains:** STAT achieves improvement on MATH by up to $7.5\%$, whereas naive fine-tuning yields negligible gains. Previous embedding-based data selection strategies that adapt to the student's validation errors are found ineffective (Section 3.2).

2. **Strong out-of-distribution (OOD) generalization:** Improvements in difficult and OOD benchmarks such as AIME24/25 and AMC23 highlight the general utility of skill-targeted training (Section 3.2).

3. **Adaptivity to evolving tasks:** Extending the previous observation, we show that STAT-Sel and STAT-Syn can be continually adapted to new, harder evaluation settings, i.e., new validation set, while still leveraging the same training set (Section 3.3).

4. **Supplementary benefits over reinforcement-learning (RL):** We show that STAT followed by RL improves upon RL-only training, such as GRPO (Shah et al., 2024) (Section 3.2). This suggests that STAT is likely to prove relevant to most training pipelines today.

We conduct extensive ablations to pinpoint the success of our proposed methods. A fine-grained skill-level analysis reveals that even when models have been extensively trained on MATH, they struggle on basic operation skills such as basic algebra (Section 4). By explicitly targeting these basic skills, our methods reduce such errors and improve generation performance, including on out-of-distribution tasks. In contrast, alternative approaches such as embedding-based methods often underperform, as they do not explicitly address these skill gaps (Figure 2). Thus, our findings demonstrate the robustness and broad applicability of skill-aware targeted training.

## 2  STAT: ADAPTING TRAINING TO MODEL'S MISSING SKILLS

Let $\mathcal{Q}$ be the set of test-set questions, out of which we use a subset $\mathcal{Q}^{val}$ as validation data and $\mathcal{Q}^{test} = \mathcal{Q} \backslash \mathcal{Q}^{val}$ as evaluation data. We also have access to a set of training questions $\mathcal{P}$, which has been utilized to train the student model during its pre-training or post-training phase, and naively fine-tuning the model on $\mathcal{P}$ offers little to no improvement. In our experiments, we use the test and training dataset from MATH, denoted as $\mathcal{Q}$ and $\mathcal{P}$ respectively. We aim to build a targeted training dataset $\mathcal{P}_{targeted}$ to train the model further.

Our work builds on using metacognitive abilities of frontier models from Didolkar et al. (2024), which we describe here. While hard to define precisely, a skill is informally defined as a basic computation necessary to solve a task at hand. For example, necessary skills to solve arithmetic tasks could be addition, subtraction, multiplication and division. We will use $\mathcal{S}$, a set of skills that are necessary to solve questions in $\mathcal{Q}$ and $\mathcal{P}$. These skills are enlisted from a large model like GPT-4o using an appropriate prompting strategy (Didolkar et al., 2024; Kaur et al., 2024). Then, we create `Skill-Map`: $\mathcal{S} \rightarrow \mathcal{P}$ to be a map from a skill to the set of training questions that require applying the skill, which we will also get by prompting the same LLM (Achiam et al., 2023). We use the skill set $\mathcal{S}$ and the `Skill-Map` from Didolkar et al. (2024).

To develop STAT-Sel and STAT-Syn, we first identify difficult questions for the student model on a validation set by analyzing its own responses. For these questions, we then use the teacher model to infer the skills that are missing from the student's reasoning. A skill-targeted training set is constructed by emphasizing examples corresponding to these missing skills, either via up-weighting samples or synthesizing new ones. Unless otherwise specified, all of our experiments use GPT-4o-mini as the teacher model.

### 2.1  STAGE 1: DETECTION OF *difficult* QUESTIONS VIA REWARD FILTERING

In this stage, we will label a question $q \in \mathcal{Q}$ as *difficult* or not for the student model. We could simply define *difficult* questions as the set of questions that the model gets wrong after evaluation. However, this requires access to the ground truth labels. Instead, to make our technique more broadly

applicable, we use a reward model to classify the responses of the student model. The reward model need not be a perfect reward model; we give more ablations in Section E.1. Given a question $q$, we use a reward model to score the response of the student model.

**Reward filtering.** As we primarily focus on math datasets, we assume that the model's response is composed of $t$ steps for a question $q$ and contains the answer in its final step. We will use the reward model to output reward scores for each step. For simplicity, we will refer to the scores of the reward model as $\{r_{q,1}, \cdots, r_{q,t}\}$. Then, we use thresholds $\tau_1, \tau_2$ to filter out *difficult* questions for the student model. We will refer to the threshold filtering function as $R : \mathcal{Q} \rightarrow \{0, 1\}$.

$$R(q) = \begin{cases} 0, & \text{(if) } r_{q,t} \leq \tau_1 & \text{(final step has low reward)} \\ & \text{(or) } \dfrac{1}{t}\sum_{i=1}^{t} r_{q,i} \leq \tau_1 & \text{(average low reward across all steps)} \\ & \text{(or) } \exists i < t \text{ s.t. } r_{q,i} \leq \tau_2 & \text{(low reward at any step)} \\ 1, & \text{otherwise,} \end{cases} \tag{1}$$

**Identifying difficult questions.** We define $\mathcal{Q}_{\text{difficult}}$ as a model-specific subset of the MATH dataset, consisting of questions with low-reward model responses $R$. To avoid training directly on the test data, we use two splits of $\mathcal{Q}_{\text{difficult}}$ as validation and test sets:

- $\mathcal{Q}_{\text{difficult}}^{\textbf{val}}$: Difficult questions in the validation set, given by $\mathcal{Q}_{\text{difficult}} \cap \mathcal{Q}^{\textbf{val}}$, are used to label missing skills in Stage 2.

- $\mathcal{Q}_{\text{difficult}}^{\textbf{test}}$: Difficult questions in the test set, given by $\mathcal{Q}_{\text{difficult}} \cap \mathcal{Q}^{\textbf{test}}$, are used for MATH$^D$ evaluation in Table 3.

### 2.2 STAGE 2: CONSTRUCTING MODEL-SPECIFIC MISSING-SKILL-PROFILE

For each *difficult* question $q$ in $\mathcal{Q}_{\text{difficult}}^{val}$, we use a frontier LLM (GPT-4o-mini) to predict the set of skills in $\mathcal{S}$ that are missing in the model's responses. We call this map `Missing-Skill-Profile`: $\mathcal{Q}_{\text{difficult}}^{val} \rightarrow \mathcal{S}$. This map will be used to build STAT-Sel and STAT-Syn. See Section 4 for examples and an extensive analysis of `Missing-Skill-Profile` across models, and Section C.3 for prompts.

### 2.3 STAGE 3: SELECTING OR SYNTHESIZING SKILL-BASED TRAINING DATA

In this stage, we construct our skill-targeted training dataset, $\mathcal{P}_{targeted}$.

**STAT-Sel.** We create this set by directly sampling questions from the training dataset $\mathcal{P}$ according to the skills listed in the `Missing-Skill-Profile`. Specifically, for each question $q \in \mathcal{Q}_{\text{difficult}}^{val}$, we examine `Missing-Skill-Profile`(q) and, for every skill it contains, sample multiple questions from $\mathcal{P}$ that are linked to the same skill via the `Skill-Map`. Consequently, the frequency with which a skill contributes to the selection process is proportional to the number of questions associated with that skill in the `Missing-Skill-Profile`.

**STAT-Syn.** We generate new synthetic questions using the teacher model. For each question $q \in \mathcal{Q}_{\text{difficult}}^{val}$, we examine `Missing-Skill-Profile`(q). For each skill it contains, we randomly sample 3 questions from $\mathcal{P}$ that are linked to the same skill via the `Skill-Map`, and ask the teacher model to create new questions and responses by referring to the sampled questions. We keep only those questions where the teacher model is consistent across at least 2 of its responses, and keep only those question-answer pairs in our training set. Detailed procedures are given in Section B.

## 3 EXPERIMENTS

### 3.1 EXPERIMENTAL SETUP

**Datasets.** All training data for STAT and the baselines are either selected or synthesized from the MATH dataset (7.5k train / 5k test) (Hendrycks et al., 2021). In addition to the original solutions

| Method | # Unique Questions | # QA Pairs | Synthetic Data | Training Data Description |
|---|---|---|---|---|
| MATH-Train (Hendrycks et al., 2021) | 7.5k | 7.5k | ✗ | MATH original training set. |
| MATH-Augment (TIGER-Lab, 2024) | 7.5k | 9.5k | ✗ | Augmented MATH training set with multiple teacher-rewritten solutions per question. |
| MATH-Hard (Sun et al., 2024) | 3k | 9.5k | ✗ | Subset of MATH-Augment with Level 4–5 MATH questions. |
| Embed-Sel (Li et al., 2025) | 4k | 9.5k | ✗ | Reweighted set of MATH-Augment via upweighting training questions similar to the *difficult* questions in embedding space. |
| Embed-Syn (Jung et al., 2025) | 4k | 9.5k | ✓ | Synthetic MATH-level QAs generated by the teacher, using training examples from Embed-Sel as references. |
| **STAT-Sel** (Ours) | 4k | 9.5k | ✗ | Reweighted set of MATH-Augment via upweighting training questions related to model's missing skills in solving the *difficult* questions. |
| **STAT-Syn** (Ours) | 4k | 9.5k | ✓ | Synthetic MATH-level QAs generated by the teacher, with training examples from STAT-Sel and their associated skills as references. |

Table 2: Comparison of training data construction methods. We attach details of data construction procedure in Section C.2.

provided in the dataset, we also collect three alternative versions of each answer by prompting the teacher model to re-write them three times. We further report performance of STAT and each baseline after continuing training with GRPO (Shao et al., 2024) on the same MATH training set. We randomly split the MATH test set into 1k validation and 4k test subsets, with both MATH and MATH$^D$ evaluations drawn from the 4k test split. See Section 2.1 for design details.

We also evaluate our method on extensive OOD benchmarks including GSM8K (Cobbe et al., 2021), MATH$^2$ (Shah et al., 2024), MATH-perturb (Huang et al., 2025), AMC23 (AI-MO, 2025), and AIME2024/2025 (HuggingFaceH4, n.d.; HuggingFaceH5, n.d.).

**Model & Training Configuration.** We focus on smaller models as a testbed, as their performance remains noticeably weaker on MATH. We employ GPT-4o-mini as the teacher model, and apply STAT on student models Llama-3.2-3B-Instruct, Llama-3.2-1B-Instruct (Meta AI, 2024), and Qwen2.5-3B (Qwen et al., 2025), and evaluate under 0-shot settings. We fine-tune models for 3 epochs, with learning rate chosen separately for each method based on accuracy on MATH. We provide detailed hyperparameters in Section C.1, ablations on threshold sensitivity in Section E.1, and a discussion of teacher model variants in Section E.3.

**Baselines.** We compare skill-aware training against several baselines. We begin with MATH-Train, where the model simply trains on the original MATH responses, and MATH-Augment, which substitutes the responses with teacher re-written answers. We also compare against MATH-Hard, restricting training to only Level 4–5 questions. Finally, to test whether skills really matter in STAT-Sel and STAT-Syn, we swap them out for an embedding-based approach[2], selecting training questions by their similarity to difficult validation questions from Stage 1. Please find a summary in Table 2. We have attached detailed data creation procedure in Section C.2 and prompts in Section C.3.

### 3.2 EVALUATION RESULTS

We present results for Llama-3.2-3B-Instruct and Qwen2.5-3B in Table 3 and for Llama-3.2-1B-Instruct in Table 7, Section D. We refer to each untrained model as 'Base Model'. Our findings can be summarized as follows.

**Naive SFT provides little to no benefit.** Both MATH-Train and MATH-Augment yield at most a 1–2% gain over the base model, showing that naive SFT offers negligible improvements. It is worth noting that we have systematically tuned hyper-parameters for naive SFT (details attached in Section C.1). In fact, we observe that Qwen2.5-3B can even degrade under MATH-Train. Restricting

---

[2]We use Alibaba-NLP/gte-Qwen2-7B-instruct (Li et al., 2023)

| Methods | MATH | MATH$^D$ | MATH$^2$ | GSM8K | AMC23 | MATH-perturb | | AIME | | Avg. |
|---|---|---|---|---|---|---|---|---|---|---|
| | | | | | | simple | hard | 2024 | 2025 | |
| *Llama-3.2-3B-Instruct + SFT* | | | | | | | | | | |
| Base Model | 44.0 | 18.2 | 21.9 | 73.0 | 21.7 | 33.7 | 12.2 | 33.3 | 16.7 | 30.5 |
| MATH-Train | 44.8 | 22.9 | 21.0 | 75.1 | 20.8 | 33.0 | 12.2 | 30.0 | 20.0 | 31.1 |
| MATH-Augment | 45.2 | 23.9 | 23.8 | 77.8 | 23.8 | 35.1 | 12.5 | 30.0 | 13.3 | 31.7 |
| MATH-Hard | 45.6 | 24.9 | 23.3 | 78.2 | 21.6 | 38.0 | 11.8 | 30.0 | 26.7 | 33.3 |
| Embed-Sel | 46.0 | 26.5 | 20.5 | 76.6 | 21.6 | 36.2 | **14.7** | 36.7 | 16.7 | 32.8 |
| Embed-Syn | 48.8 | 27.3 | 19.5 | 78.4 | 22.7 | 36.9 | 13.3 | 26.7 | 23.3 | 33.0 |
| **STAT-Sel** | **51.5** | 26.6 | 25.7 | **80.2** | **24.7** | 39.8 | 13.3 | 43.3 | 23.3 | 36.5 |
| **STAT-Syn** | 50.2 | **31.7** | **26.2** | 79.2 | 23.9 | 39.1 | **14.7** | 40.0 | **30.0** | **37.2** |
| *+ GRPO* | | | | | | | | | | |
| Base Model | 45.4 | 24.4 | 23.3 | 77.4 | 25.8 | 38.4 | 11.8 | 33.3 | 3.3 | 31.8 |
| MATH-Train | 46.4 | 28.4 | 28.6 | 80.7 | 29.7 | 37.6 | 12.5 | 36.7 | 10.0 | 34.5 |
| MATH-Augment | 47.4 | 31.6 | 28.6 | 81.4 | 30.6 | 37.6 | 14.0 | 36.7 | **33.3** | 37.9 |
| MATH-Hard | 49.4 | 33.2 | 28.6 | 80.3 | 31.3 | 39.1 | 15.4 | 43.3 | 13.3 | 37.1 |
| Embed-Sel | 50.4 | 37.5 | 23.8 | 80.5 | 32.0 | 38.0 | 16.8 | 36.7 | 20.0 | 38.8 |
| Embed-Syn | 49.7 | 37.8 | 19.5 | 80.6 | 33.9 | 39.1 | 16.8 | 36.7 | 23.3 | 38.6 |
| **STAT-Sel** | **52.2** | 35.0 | **32.4** | 81.8 | **34.2** | 42.7 | **17.6** | 43.3 | 26.7 | 40.7 |
| **STAT-Syn** | 51.0 | **39.1** | 29.0 | **82.0** | 31.9 | **43.0** | 15.8 | **46.7** | **33.3** | **41.3** |
| *Qwen2.5-3B + SFT* | | | | | | | | | | |
| Base model | 55.8 | 45.3 | 34.8 | 80.9 | 26.4 | 43.7 | 24.0 | 23.3 | 20.0 | 39.4 |
| MATH-Train | 50.0 | 44.2 | 32.9 | 80.1 | 33.6 | 42.3 | 23.3 | 26.7 | 26.7 | 40.0 |
| MATH-Augment | 56.6 | 45.6 | 37.1 | 80.4 | 33.0 | 40.9 | 21.9 | 16.7 | 26.7 | 39.9 |
| MATH-Hard | 56.7 | 45.6 | 31.4 | 79.8 | 33.6 | 43.7 | 23.7 | 30.0 | 16.7 | 40.1 |
| Embed-Sel | 57.5 | 46.4 | 34.3 | 80.4 | 33.6 | 43.7 | 21.9 | 30.0 | 26.7 | 41.6 |
| Embed-Syn | 56.4 | 47.4 | 34.3 | 80.4 | 35.2 | 43.7 | 24.0 | 26.7 | 26.7 | 41.6 |
| **STAT-Sel** | 58.4 | 47.6 | 39.5 | **82.3** | 35.5 | 45.9 | 24.0 | 33.3 | 30.0 | 44.1 |
| **STAT-Syn** | **59.4** | **49.2** | **40.5** | 81.3 | 34.4 | 44.8 | **25.1** | **36.7** | 30.0 | **44.6** |
| *+ GRPO* | | | | | | | | | | |
| Base model | 61.6 | 49.8 | 41.0 | 85.1 | 37.7 | 49.8 | 25.8 | 33.3 | 30.0 | 46.0 |
| MATH-Train | 61.6 | 51.1 | 34.8 | 84.8 | 36.9 | **51.6** | 26.5 | 33.3 | 30.0 | 45.6 |
| MATH-Augment | 61.0 | 48.2 | 40.5 | 84.0 | 36.3 | 48.7 | 26.2 | 36.7 | 26.7 | 45.4 |
| MATH-Hard | 59.0 | 51.1 | 35.7 | 84.2 | 37.7 | 49.8 | 26.5 | 33.3 | 23.3 | 44.5 |
| Embed-Sel | 59.7 | 48.9 | 41.0 | 84.3 | 38.4 | 46.6 | 25.8 | 26.7 | **36.7** | 45.3 |
| Embed-Syn | 61.4 | 52.3 | 40.0 | 83.7 | 38.8 | 47.7 | 28.0 | 26.7 | 30.0 | 45.4 |
| **STAT-Sel** | **62.8** | 52.1 | **44.8** | 84.8 | 38.8 | 48.7 | **30.1** | 36.7 | 33.3 | 48.0 |
| **STAT-Syn** | 61.8 | 52.4 | 41.9 | **85.6** | 39.2 | 50.9 | 26.9 | **40.0** | **36.7** | **48.4** |

Table 3: Improvements on various math benchmarks from applying STAT. Results under '+SFT' show the performance of SFT models trained with each method, while '+GRPO' shows the performance after applying GRPO on top of the corresponding SFT models. Our methods, STAT-Sel and STAT-Syn, achieve an average gain of up to 6.7% over the base model, with strong OOD performances (AMC23 results reported on average@64, AIME on pass@64). Applying GRPO on top of fine-tuning with STAT further enhances these improvements by ~4%. Full results are provided for Llama-3.2-1B-Instruct in Table 7, Section D.

supervision to only the most difficult MATH questions (Levels 4–5) also fails to produce meaningful gains. A natural idea is then to adapt training toward the model's mistakes by selecting training questions semantically close to difficult validation examples. Using embedding similarity, Embed-Sel achieves only marginal over MATH-Train and MATH-Augment. Synthetic augmentation via Embed-Syn provides a small additional boost, but the overall gains remain modest.

**Skill-targeted adaptive training shows substantial improvements.** STAT achieves average gains of up to 6.7% on Llama-3.2-3B-Instruct, 5.2% on Qwen2.5-3B, and 3.4% on Llama-3.2-1B-Instruct, over the performance of base model. On closer analysis on MATH$^D$ test set of questions, we show that STAT-Syn substantially improves the performance of the model on difficult questions, compared to STAT-Sel, which leads to improved performance overall for Llama-3.2-1B-Instruct and Qwen2.5-3B.

**Benefits extend beyond MATH.** On out-of-distribution benchmarks, we observe consistent improvements across 7 datasets, ranging from simpler problems in GSM8K to challenging competition sets

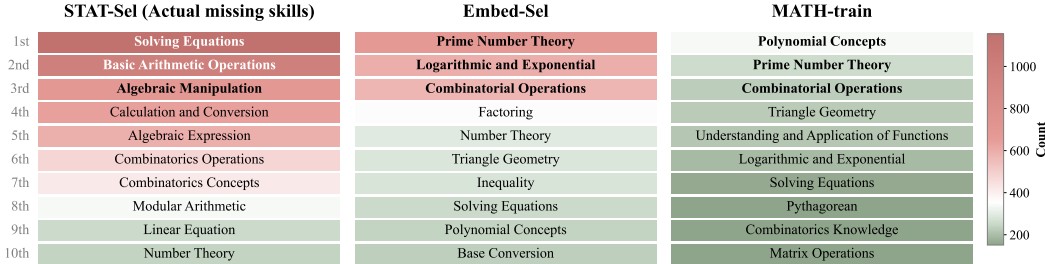

Figure 2: Comparison among the Top 10 frequent skills present in STAT-Sel, Embed-Sel, and MATH-Train questions selected on Llama-3.2-1B-Instruct. The skills emphasized in both baselines, MATH-Train and Embed-Sel, align poorly with the actual Top 10 missing skills of the model (i.e., skills in STAT-Sel).

such as AIME. Specifically, STAT-Sel and STAT-Syn improve averaged OOD performances by $5.3\%$ and $5.8\%$ respectively, with STAT-Syn generally excelling on harder tasks such as AIME and MATH$^D$. This demonstrates that targeting skills generalizes extensively beyond the source training set.

**Compatibility with GRPO.** A natural concern is whether our methods can work well with RL-based methods such as GRPO, which typically follows SFT (Dubey et al., 2024; Guo et al., 2025). For both Llama and Qwen, improvements from SFT on STAT have carried over to subsequent GRPO, yielding average gains of up to $9.5\%$ over GRPO on base model. Surprisingly, on Llama-3.2-1B-Instruct and Llama-3.2-3B-Instruct, where GRPO alone does not work well (improving ≤$2.4\%$), SFT alone on STAT already produced better results than GRPO, and adding GRPO on top further boosts performance by ~$4\%$.

### 3.3 CONTINUAL LEARNING ON CHALLENGING BENCHMARKS

As our earlier results show, STAT already generalizes strongly to a wide range of OOD tasks while using only MATH data for training. But in practice, models often face evaluation settings that grow harder over time. A natural question then is: *can we continue adapting the model to these tougher benchmarks while still using similar questions as MATH?*

For our case study, we consider MATH-perturb-hard. We report performance for two model variants of Llama-3.2-3B-Instruct. **STAT-ConSel** takes a model trained with STAT-Sel, and trains further with a data creation pipeline identical to STAT-Sel, but with `Missing-Skill-Profile` built on validation questions from MATH-perturb-hard. **STAT-ConSyn** builds on STAT-Syn model with the same idea. In both cases, the evaluation benchmark only gives the skill profile, and the training examples still come from MATH.

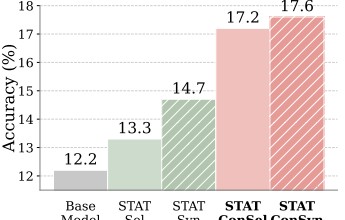

Figure 3: Continual learning results on MATH-perturb-hard. Further fine-tuning STAT models based on their missing skills on unseen data yields a 3–4% gain (STAT-ConSel/ConSyn).

As shown in Figure 3, STAT-Sel and STAT-Syn trained models show only 1–2% improvement on MATH-perturb-hard over the base model performance, which reflects the difficulty of this benchmark. However, continual trained models show a larger gain of 3–4%. This shows that our framework can be readily adapted to unseen test-time datasets by constructing `Missing-Skill-Profile` directly on them, while still using MATH training data. Thus, skill-aware training provides a flexible solution to adapt the models with more challenging evaluations while still relying on existing training datasets.

## 4 WHY SKILL-TARGETED TRAINING WORKS

In this section, we dig into the effectiveness of our proposed skill-aware targeted training. We conduct all the ablations and analyses on Llama-3.2-1B-Instruct due to limited computational resources. First, we present the `Missing-Skill-Profile` across all models. We then show that STAT improves the student's performance uniformly across these skills. Finally, we show that the baseline strategies are ineffective because of misalignment in the skill distribution in their proposed training data and the missing skills.

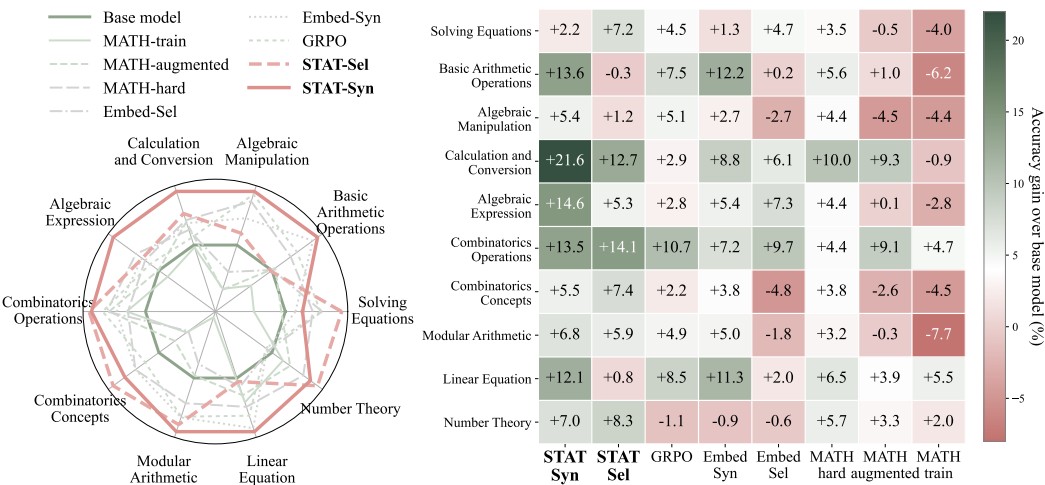

Figure 4: Trained model performances **(Left)** and performance gain over base model **(Right)** on Top 10 frequent missing skills, across training strategies on Llama-3.2-1B-Instruct. Accuracies on the left plot are normalized per skill axis for better visualization. Our approaches STAT-Syn and STAT-Sel are most effective in enhancing model performance across nearly all the skills.

**Models struggle with basic skills.** First, we closely examine the `Missing-Skill-Profile` across different models, obtained at the end of Stage 2 (Section 2.2). We present the Top 10 frequently missing skills for each model according to their `Missing-Skill-Profile` in Figure 2 (Left) and Figure 6 (appendix D). The key observations are:

• **Algebra-centric skills appear at the top,** e.g., manipulating equations, handling expressions, and solving linear forms. This suggests that even though both Llama and Qwen models achieve high performance on MATH, they systematically struggle with operation computations.

• **Most missing skills are shared across models,** e.g., equation-solving skills and basic arithmetic operations are missing in different model families (Llama and Qwen) and sizes (1B and 3B). However, smaller models show more frequent weaknesses in basic computational skills like arithmetic.

**STAT effectively addresses models' frequent missing skills:** We take Llama-3.2-1B-Instruct as a case study to examine how different training strategies impact performance across skills. From its `Missing-Skill-Profile`, we select the 10 most frequently missing skills and build corresponding evaluation sets, each containing questions annotated via the `Skill-Map`. We then measure both absolute performance and performance gains under each method.

As shown in Figure 4 (Left), STAT consistently outperform all baselines across all 10 skills, whereas baseline models can even fall behind the base model on skills such as Algebraic Manipulation and Modular Arithmetic. Figure 4 (Right) provides a quantitative breakdown, showing that STAT can deliver over 10% accuracy gains on 5 skills, with the largest improvements on basic skills like Calculation & Conversion, Algebraic Expression, and Combinatoric Expressions. Notably, STAT also brings clear improvements on knowledge-intensive skills such as Number Theory and Combinatorics.

**Misalignment between baseline training data and missing skills.** To investigate the reason behind the ineffectiveness of our baseline strategies, we adopt a skill-based evaluation by comparing the skill distribution of their training data with the model's missing skills in the `Missing-Skill-Profile`. Figure 2 highlights a clear misalignment between the model's actual missing skills (STAT-Sel) and the baselines: Neither MATH-Train nor Embed-Sel targeted the basic algebraic skills the model struggles with, even though Embed-Sel selects data directly via embedding similarity to difficult questions. They prioritize more advanced and conceptual areas such as polynomials, prime number theory, and trigonometric or matrix operations. We provide concrete question examples in Section D.3 to illustrate the distinct emphasis of each skill.

**Comparing STAT to GRPO.** One of our interesting findings in Section 3.2 was that STAT could outperform GRPO training on Llama instruct models. Here, we compare these two approaches from a skill-based perspective. As shown in Figure 4 (Right), although GRPO on Llama-3.2-1B-Instruct also yields positive gains across nearly all the top skills, the overall effect remains less pronounced compared to STAT. A possible reason is that GRPO provides only coarse feedback to the model by contrasting correct and incorrect responses, whereas skill-targeted training pinpoints model weaknesses in a fine-grained way. In light of this, one future direction is to develop a GRPO variant that incorporates skill-based feedback into the reward.

**Case study on synthetic data.** To understand why our training samples are skill-targeted, we conduct a case study of the training data.

Here we compare STAT-Syn with Embed-Syn, as their data are both created with a specific focus (e.g., embedding-based similarity or missing-skill targeting).

> **Original Question (on Ellipse Properties)**
>
> Let $F_1$ and $F_2$ be the foci of the ellipse $kx^2 + y^2 = 1$, where $k > 1$ is a constant. Suppose that there is a circle which passes through $F_1$ and $F_2$ and which lies tangent to the ellipse at two points on the $x$-axis. Compute $k$.

> **Model Response & Missing Skill (on Solving Equations)**
>
> We can rewrite in the standard form of an ellipse: $\frac{x^2}{1^2} + \frac{y^2}{(1/\sqrt{k})^2} = 1$. We can see that the length of the semi-horizontal axis is 1 and the semi-vertical axis is $\frac{1}{\sqrt{k}}$. **(Correct ✓)**
> ......
> Therefore, we have $\frac{\sqrt{k-1}}{\sqrt{k}} = \frac{1}{\sqrt{k}}$. Simplifying this equation, we get: $\sqrt{\frac{1}{k} - 1} = \sqrt{\frac{1}{k}} - 1$. This equation is true for all values of $k$. Therefore, the value of $k$ is not uniquely determined by the given conditions. **(Incorrect ✗ , Missing skill: Solving Equations)**

> **Embed-Syn Question (on Ellipse Properties)**
>
> The ellipse $\frac{x^2}{9} + \frac{y^2}{4} = 1$ has foci located along one of the coordinate axes. What is the distance between the foci?

> **STAT-Syn Question (on Solving Equations)**
>
> Solve for $x > 0$:
> $$\frac{1}{\sqrt{x+4}} = 2.$$

Figure 5: Comparison between synthesized questions from Embed-Syn and STAT-Syn.

In this example (see Figure 5), the original question centers on ellipse geometry; the model handles this part well, but showed a gap in the final equation-solving step. The new question in Embed-Syn, though highly relevant, captures only the main topic (Ellipse Properties) through embedding similarity. By contrast, STAT-Syn leverages the missing-skill information (Solving Equations) and generates a targeted question to practice it.

This case study demonstrates that semantic similarity, as captured by embedding-based methods, is not always the right approach. Skill-targeted adaptive training provides a direct way to constantly improve the model.

## 5 DISCUSSION

**Related Works:** We provide a more detailed discussion of related works in Section A. Broadly, prior approaches can be grouped into three directions. First, several skill-aware algorithms improve language models either by designing more targeted inference-time instructions or by generating synthetic data to instill new skills (Kaur et al., 2024; Gandhi et al., 2025; Didolkar et al., 2024). In contrast, our method adapts training data toward skills that the model continues to struggle with, even after having been extensively trained.

Second, performance-aware adaptation methods adjust training data to improve efficiency and performance (Xia et al., 2024; Yu et al., 2024b; Xie et al., 2023b). However, these techniques largely focus on minimizing validation loss on a target set, which is only an indirect proxy for generation-time errors. Some attempts to remove dependence on explicit validation sets instead optimize implicit properties such as embedding or gradient diversity (Jung et al., 2025; Wang et al., 2024; Yu et al., 2024a; Ni et al., 2024). By contrast, our approach explicitly targets the model's generation mistakes through a metacognitive framework.

We also relate our work to recent approaches that adapt training by profiling model weaknesses (Zeng et al., 2025; Khan et al., 2025). These methods construct skill representations to measure student performance and guide synthetic data generation. For instance, Khan et al. (2025) study adaptive data generation agents that allocate training data across skill categories based on student performance, while Zeng et al. (2025) build hierarchical skill trees by clustering instance-level skill descriptions to generate targeted synthetic data.

While these works and ours all leverage skill-level profiling, the role of skills differs. Prior approaches use skills as organizational categories for summarizing correctness across instances and guiding synthetic data generation to those categories where models struggle. In contrast, we use skills to identify missing components in failed reasoning generations. We further show that modifying question distribution through missing skills, without generating new data, can improve performance. Our approach is complementary and provides a finer-grained strategy on adaptive training.

Finally, while existing works have shown effectiveness of teacher-in-the-loop training (Zhou & Ai, 2024; Gu et al., 2024; Zhang et al., 2024b; Wang et al., 2023; Zhou et al., 2023; Xu et al., 2024), where a teacher provides repeated feedback to student through logits or targeted generations, our skill-aware targeted training provides an effective and cheaper way to adapt student with the teacher.

**Conclusion:** We investigate whether targeted skill-based training can improve language models when naive re-training yields little benefit. Using a frontier LLM to analyze responses, we construct a skill profile and selectively re-train on relevant examples, achieving significant gains on both in- and out-of-distribution tasks. Ablations show that models often fail on basic skills like algebraic computations, and STAT efficiently addresses such gaps with carefully adapting training data.

Our work points to two promising directions for future research. First, since the general skill feedback identified by a frontier LLM can effectively guide student training, it would be valuable to investigate whether these skills correspond to specific mechanistic circuits within the model. Second, while our focus has been on mathematical datasets, exploring whether STAT can also improve dimensions such as safety and interpretability presents an interesting avenue for further study.

# 6 ETHICS STATEMENT

All authors of this work have read and agree to abide by the ICLR Code of Ethics. We affirm that this research was conducted in compliance with the principles of research integrity, fairness, and transparency outlined therein.

Our study focuses on developing and evaluating a novel fine-tuning approach for language models, targeted at improving mathematical reasoning benchmarks. The work exclusively utilizes publicly available datasets such as MATH, AMC23, and AIME24/25. These datasets are widely used in the research community and do not involve human subjects, private data, or personally identifiable information. No sensitive, proprietary, or confidential data were accessed or released.

We acknowledge that research in language model training can have broader societal impacts, particularly regarding potential misuse (e.g., generating misleading or harmful content). However, our contributions are specifically focused on mathematical problem-solving and skill-targeted fine-tuning, which pose minimal direct risk of harmful applications. The methods proposed are not designed for, nor evaluated on, domains involving sensitive personal, social, or political content.

We have no conflicts of interest or external sponsorships that could bias the reported results. All experiments were performed under standard academic conditions with openly available resources. Our work complies with legal and ethical standards for dataset usage, algorithm development, and reporting.

**Use of LLM:** We used an LLM solely to improve the clarity and readability of the manuscript text (e.g., grammar and style polishing). The model was not employed for designing experiments, analyzing data, or generating results. All scientific contributions, methodologies, and findings reported in this work are the product of the authors.

**Reproducibility Statement:** We have taken several steps to ensure the reproducibility of our results. A detailed description of the STAT algorithms, including pseudocode, is provided in Section 2 and Section B.1. The datasets used in all experiments (MATH, AMC23, AIME24/25, GSM8K, and others) are publicly available and fully cited in the references. We describe our experimental setup, model configurations, training hyperparameters, and ablations in Section 3.1 and Section C.

To facilitate replication, we will release our code repository along with all the STAT-Syn data if we proceed to the camera-ready version. Together, these resources provide sufficient detail for independent researchers to reproduce our results and extend our methods to related benchmarks.

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

## A    RELATED WORKS

Recent line of works have shown that cognitive theories relevant for human learning can also be utilized to improve performance of language models. Arora & Goyal (2023) argue that language models learn to generalize beyond training data, by learning generalizable skills that connect the text tokens. This argument was later used by Wu et al. (2024); Yu et al. (2023a); Zhao et al. (2024) to create evaluation benchmarks to compare how different LLMs can generalize beyond training. Furthermore, Didolkar et al. (2024); He et al. (2025) utilize this framework to create effective instance specific in-context learning examples for language models. More related to our work is the work by Kaur et al. (2024) who create synthetic instruction following datasets using arbitrary combination of skills and show that small language models learn more efficiently from such synthetic tasks. Similarly, Gandhi et al. (2025) show that certain cognitive skills are necessary for models to explore during reinforcement learning, and one can similarly enforce such skills by targeted continual pretraining. In contrast, we show that we can use the skill-based framework to create targeted training datasets by analyzing the missing skills in model's responses after training and even unlock further gains.

Related to our work is a line of research that leverages model gradients to construct targeted training datasets (Xia et al., 2024; Yu et al., 2024b). These approaches utilize gradients on both training and test sets to identify effective subsets of training data. However, a key limitation is their reliance on ground-truth solutions for the test data. In contrast, our work offers a complementary strategy: we employ a frontier LLM to analyze the model's responses and generate targeted training data based on its feedback.

We also compare our proposed method to embedding based data selection. Embedding based methods Wang et al. (2024); Yu et al. (2024a); Ni et al. (2024) have primarily been used to improve diversity of training dataset. As shown by our results, embedding similarity could also be utilized (albeit with a drop in performance) to get questions from training dataset that are similar to difficult questions.

Finally, we introduce STAT-Syn, an approach analogous to STAT-Sel, which synthesizes additional training data specifically targeted to the identified missing skills. Synthetic data generation has recently attracted significant attention as a practical alternative for augmenting real-world datasets, and has been shown to improve language model performance both in-distribution and out-of-distribution (Jung et al., 2025; Yu et al., 2023b; Lu et al., 2023; Li et al., 2023; Kaur et al., 2024). A comprehensive comparison of STAT-Syn with prior work on synthetic data generation is left to future study.

## B    DETAILS OF STAT DATA CREATION

### B.1    ALGORITHM FOR CONSTRUCTING STAT-SEL AND STAT-SYN DATA

Algorithm 1 outlines the procedure to construct $\mathcal{P}_{targeted}$ in Stage 3 (Section 2). For each question in the test set $\mathcal{Q}$, the algorithm first identifies the associated missing skills using the Missing-Skill Profile. For each missing skill, a small set of examples is retrieved from the Skill-Map, which links each skill to corresponding training data. In STAT-Sel, these retrieved examples are directly added to the target training set. Otherwise, the examples are used as seeds to prompt GPT-4o to generate new, skill-specific questions, which are then included instead. This process ensures that the resulting training set $\mathcal{P}_{targeted}$ is adaptively enriched with examples that directly address the model's weaknesses.

### B.2    TRAINING DATA CREATION PROCEDURE OF STAT

We now provide a detailed interpretation of our training data creation approach outlined Algorithm 1.

**STAT-Sel.**    4k unique questions, 9.5k QA pairs. We begin by filtering 500 difficult questions from the validation set using our process reward model. For each such question, the teacher model identifies 2–3 missing skills in the student's response. As described in Section 2.3, we then create the training set by selecting 5 questions for each missing skill in the question's `Missing-Skill-Profile`. We use 3 answers for each question and randomly sample a subset of 9.5k question-answer pairs as our training set.

---

**Algorithm 1** Skill-based data selection/generation

---

**Input:** Test set $\mathcal{Q}$, `Skill-Map`: $\mathcal{S} \rightarrow \mathcal{P}$, `MissingSkillProfile`: $\mathcal{Q} \rightarrow \mathcal{S}$, STAT-Sel: bool
**Output:** $\mathcal{P}_{targeted}$

 1: $\mathcal{P}_{targeted} \leftarrow$ [ ]
 2: **for** $q$ in $\mathcal{Q}$ **do**
 3:     `skill_list` $\leftarrow$ `MissingSkillProfile[q]`
 4:     **if** `skill_list` is not empty **then**
 5:         **for** `skill` in `skill_list` **do**
 6:             $\mathcal{P}_{skill} \leftarrow$ `Skill-Map[skill]`
 7:             $\mathcal{P}_{selected} \leftarrow$ `random_sample(`$\mathcal{P}_{skill}$`, 3)`
 8:             **if** STAT-Sel **then**
 9:                 $\mathcal{P}_{targeted} \leftarrow \mathcal{P}_{targeted} + \mathcal{P}_{selected}$
10:             **else**
11:                 $\mathcal{P}_{new} \leftarrow$ GPT-4o($\mathcal{P}_{selected}$, `skill`, prompt="Propose a new question based on
12:                                             the given questions and the given skill.")
13:                 $\mathcal{P}_{targeted} \leftarrow \mathcal{P}_{targeted} + \mathcal{P}_{new}$
14:             **end if**
15:         **end for**
16:     **end if**
17: **end for**
18: **return** $\mathcal{P}_{targeted}$

---

**STAT-Syn.** 4k unique questions, 9.5k QA pairs. We begin by filtering 500 difficult questions from the validation set using our process reward model. For each such question, the teacher model identifies 2–3 missing skills in the student's response. For each pair of (`difficult_question`, `missing_skill`), we retrieve 3 questions from MATH training set. We input these 3 questions, along with the `missing_skill`, to the teacher model, prompting it to synthesize 2 new questions. The teacher further generates 3 solutions for each new question. We then filter the newly synthesized data by:

**1.** Compute consistency scores for each set of (`new_question, solution`) pairs, according to the number of solutions agreeing on the final answer. For example, a new question with 2 solutions agreeing on the final answer has a consistency score of 2.

**2.** Keep only the `new_question` with a consistency score of $\geq 2$.

**3.** For each filtered question, keep only the `solution` that agrees on the final answer. [3]

This process enables our approach to generate diverse data, as we input 3 questions to the teacher model as references each time. The consistency-filtering step filters out both invalid questions and solutions, ensuring the quality of STAT-Syn.

---

[3]For STAT-Syn, after filtering teacher-generated answers using consistency, we obtained 9.5k valid question–answer pairs. To ensure comparability, we standardize the training data size to 9.5k pairs for all experiments.

## C  Experimental details

### C.1  Model & Training Configurations

**Model Settings.** All inferences are under 0-shot settings, with temperature 0.1 for pass@1 sampling, and temperature 1.0 for average@64 or pass@64 sampling. For the process reward model in Stage 1 (Section 2.1), we use RLHFlow/Llama3.1-8B-PRM-Mistral-Data (Xiong et al. (2024)), an 8B process reward model fine-tuned from Llama-3.1-8B, with filtering thresholds $\tau_1 = 0.85, \tau_2 = 0.7$. We use seed=0 for all evaluations.

**SFT configurations.** For SFT, we adopt QLoRA with rank 16, scaling factor $\alpha = 32$, and dropout 0.05, applied to the attention and MLP projection modules. Models are trained in 4-bit NF4 quantization with bfloat16 compute, using the paged AdamW (8-bit) optimizer. We train for 3 epochs with a cosine learning rate schedule and a 3% warmup ratio. Peak learning rate is chosen separately for each method among {5e-4, 2e-4, 1e-4, 8e-5, 2e-5, based on accuracy on MATH. The effective batch size is 8 (per-device batch size of 2 with gradient accumulation of 4). We apply gradient clipping at 0.3, weight decay of 0.1, and enable group-by-length packing for efficiency. Other configurations follow the official code base from Llama[4] and Qwen[5].

**GRPO configuration.** We train for 6 epochs using a constant learning rate of 5e-7. The objective includes only the policy update loss, without any KL-divergence term, and the entropy coefficient is fixed at $0.0$. Each batch contains 256 questions, with 4 rollouts generated per question. Responses are truncated at a maximum length of 2048 tokens. We set the PPO mini-batch size to 64, which implies that each batch of 256 questions is split into four mini-batches. The model performs four gradient updates before refreshing the reference model.

### C.2  Training Data Creation Procedure of Baselines

We compare STAT-Sel and STAT-Syn with the following baseline models fine-tuned with various data selection/generation methods, to measure the effectiveness of skill-aware training:

• **MATH-Train:** 7.5k unique questions, 7.5k QA pairs. We naively train the model on all question from the training dataset, with a single answer from the original dataset for each question.

• **MATH-Augment:** 7.5k unique questions, 9.5k QA pairs. In order to make a fair comparison to our proposed methods, we pick 3 answers per question to create 22.5k question-answer pairs and then randomly sample a subset of 9.5k question answer pairs as our training set.

• **MATH-Hard:** 3k unique questions, 9.5k QA pairs. We include all questions from the levels 4 and 5 of the MATH dataset. We use 3 responses per question to create a pool of 12k question-answer pairs and then keep a random subset of 9.5k question answer pairs.

• **Embed-Sel:** 4k unique questions, 9.5k QA pairs. Here, we compare the effectiveness of skill-based training data selection to embedding-based training data selection [6]. We use our *difficult* question set from stage 1 and for each question, we pick 5 similar questions from the training set using an embedding model based similarity score. We pick 3 answers per selected questions and keep a random subset of 9.5k question answer pairs.

• **Embed-Syn:** 4k unique questions, 9.5k QA pairs. For each question in the difficult set identified during stage 1, we retrieve 5 question–answer pairs from the training set $\mathcal{P}$ using an embedding-based similarity measure. The teacher model is then prompted to generate 5 new questions, each accompanied by 3 candidate responses, conditioned on different groups of 3 retrieved pairs as in-context examples. We retain only those generated questions for which the LLM produces at least 2 consistent responses, and add the corresponding consistent question–answer pairs to our training set. Finally, we keep a random subset of 9.5k question answer pairs to create our training set.

---

[4] https://github.com/meta-llama/llama-cookbook

[5] https://github.com/QwenLM/Qwen

[6] We use Alibaba-NLP/gte-Qwen2-7B-instruct as our embedding model (Li et al., 2023)

### C.3 PROMPTS

### C.3.1 CONSTRUCTING SKILL-MAP ON MATH

**Statistics of skill lists.**   We adopt the list of mathematical skills obtained in Didolkar et al. (2024) using an LLM labeling→clustering pipeline. The skill list contains 128 skills in total, divided into 7 subsets across 7 subjects. Each subject includes ~18 skills.

**Skill-Map construction procedure.**   To construct the Skill-Map (see Section 2), we follow Didolkar et al. (2024) to label skills on both the training and test sets of MATH using GPT-4o-mini (OpenAI, 2024). We enlist all skills that we used to annotate the questions in MATH and dataset in Tables 5 and 6, which have been taken from Didolkar et al. (2024). We ask the LLM to read the question and provide up to five skills required to solve this question, from the given existing skill list. We show an example prompt for annotating MATH Number Theory questions as follows.

---

**Example skill annotation prompt for MATH Number Theory questions**

[TASK]
You'll be given a math question. Your task is to output:
(1) < skill> list here up to five skill(s) that are required to solve this problem, seperated by commas </skill>.
(2) <reason> reason here why these skills are needed </reason>.

[SKILL LIST]
You should only choose the skills from this list:
[
"arithmetic_sequences",
"base_conversion",
"basic_arithmetic",
"division_and_remainders",
"exponentiation",
"factorization",
"greatest_common_divisor_calculations",
"modular_arithmetic",
"number_manipulation",
"number_theory",
"polynomial_operations",
"prime_number_theory",
"sequence_analysis",
"solving_equations",
"understanding_of_fractions"
]

[QUESTION]
{question}

[REASON AND SKILL(S)]

---

Table 4 shows some example MATH questions and their corresponding annotated skills. From the skill annotation, we construct a Skill-Map (see Section 2) that stores the required skills for each question.

| Question | Annotated skills |
|---|---|
| What is the units digit of $3^1 + 3^3 + 3^5 + 3^7 + \ldots + 3^{2009}$? | exponentiation, modular arithmetic, sequence analysis |
| In the addition problem each letter represents a distinct digit. What is the numerical value of E? [Figure] | basic arithmetic, number manipulation, solving equations |
| In triangle $ABC$, $\tan(\angle CAB) = \frac{22}{7}$, and the altitude from $A$ divides $\overline{BC}$ into segments of length 3 and 17. What is the area of triangle $ABC$? | geometry and space calculation, trigonometric calculations, arithmetic operations |

Table 4: Example MATH questions, and the annotated skills generated by GPT-4o-mini.

| Subject | List of Skills |
|---|---|
| Per subject split in MATH | |
| Algebra | `algebraic_expression_skills`, `algebraic_manipulation_skills`, `arithmetic_skills`, `calculation_and_conversion_skills`, `combinatorial_operations_and_basic_arithmetic`, `complex_number_skills`, `distance_and_midpoint_skills`, `exponent_and_root_skills`, `factoring_skills`, `function_composition_skills`, `function_skills`, `geometric_sequence_skills`, `graph_and_geometry_skills`, `inequality_skills`, `logarithmic_and_exponential_skills`, `number_theory_skills`, `polynomial_concepts`, `quadratic_equation_skills`, `ratio_and_proportion_skills`, `sequence_and_series_skills`, `solving_equations` |
| Counting and Probability | `calculating_and_understanding_combinations`, `combinatorial_mathematics`, `combinatorics_concepts`, `counting_principals`, `factorials_and_prime_factorization`, `number_theory_and_arithmetic_operations`, `permutation_and_combinations`, `probability_calculation_with_replacement`, `probability_concepts_and_calculations`, `probability_theory_and_distribution`, `combinatorics_operations` |
| Geometry | `3d_geometry_and_volume_calculation_skills`, `algebraic_skills`, `area_calculation_skills`, `circle_geometry_skills`, `combinatorics_and_probability_skills`, `coordinate_geometry_and_transformation_skills`, `other_geometric_skills`, `pythagorean_skills`, `quadrilateral_and_polygon_skills`, `ratio_and_proportion_skills`, `triangle_geometry_skills`, `trigonometry_skills`, `understanding_circle_properties_and_algebraic_manipulation` |

Table 5: List of skills used for annotating questions in each subject in MATH dataset

| Subject | List of Skills |
|---|---|
| Per subject split in MATH | |
| Intermediate Algebra | `absolute_value_skills,` `algebraic_manipulation_and_equations,` `calculus_optimization_skills,` `complex_number_manipulation_and_operations,` `function_composition_and_transformation,` `graph_understanding_and_interpretation,` `inequality_solving_and_understanding,` `polynomial_concepts,` `properties_and_application_of_exponents,` `quadratic_equations_and_solutions,` `recursive_functions_and_sequences,` `sequence_and_series_analysis_skills,` `simplification_and_basic_operations,` `solving_inequalities, solving_system_of_equations,` `summation_and_analysis_of_series,` `understanding_and_application_of_functions,` `understanding_and_applying_floor_and_ceiling_functions,` `understanding_and_manipulation_of_rational_functions,` `understanding_and_utilizing_infininte_series,` `understanding_ellipse_properties,` `understanding_logarithmic_properties_and_solving_equations` |
| Number Theory | `arithmetic_sequences, base_conversion,` `basic_arithmetic, division_and_remainders,` `exponentiation, factorization,` `greatest_common_divisor_calculations,` `modular_arithmetic, number_manipulation,` `number_theory, polynomial_operations,` `prime_number_theory, sequence_analysis,` `solving_equations, understanding_of_fractions` |
| Pre-algebra | `average_calculations, basic_arithmetic_operations,` `circles, counting_and_number_theory,` `exponentiation_rules, fractions_and_decimals,` `geometry, multiples_and_zero_properties,` `multiplication_and_division, perimeter_and_area,` `prime_number_theory, probability_and_combinatorics,` `ratio_and_proportion, linear_equation` |
| Pre-calculus | `algebra_and_equations, basic_trigonometry,` `calculus, complex_number_operations,` `complex_numbers, coordinate_systems,` `determinant_calculation, geometric_relations,` `geometry_and_space_calculation,` `geometry_triangle_properties, matrix_operations,` `parametric_equations,` `sequences_series_and_summation,` `three_dimensional_geometry,` `trigonometric_calculations, vector_operations` |

Table 6: List of skills used for annotating questions in each subject of MATH dataset (continued from Table 5)

### C.3.2 Missing skill Identification from Model Responses

In Stage 2 of STAT, we use GPT-4o-mini to label the skills that are missing from a model response. We ask the LLM to read the question along with the SLM response and provide the skills that the model fails to leverage in the response, from the given existing skill list. Below we show an example prompt for labeling missing skills for MATH Number Theory questions, as well as an example LLM output.

---

**Example missing skill identification prompt for a MATH Number Theory question**

[TASK]
You'll be given a math question and a step-by-step solution written by a Small Language Model. Your task is to output:
(1) <judge> judge here whether the solution is correct or incorrect </judge>
(2) <reason> if it's incorrect, reason here why the solution is incorrect </reason>,
(3) <skill> list here what skill(s) should the SLM enhance in order to answer correctly, seperated by commas </skill>.

[SKILL LIST]
{list of 15 number theory skills}

[QUESTION]
{question}

[MODEL_SOLUTION]
{model_response}

[YOUR REASON AND SKILL(S)]

---

**Example GPT-4o-mini output for missing skill identification**

(1) <judge> incorrect </judge>
(2) <reason> The solution incorrectly simplifies the expression and misapplies properties of modular arithmetic. The solution claims that the sum of the inverses from 1 to ( p-1 ) is (-1 pmodp), which is incorrect. The correct approach involves recognizing that each term $\left(a^{-1} \cdot (a+1)^{-1}\right)$ should be evaluated individually, and the sum should be calculated directly without assuming incorrect simplifications. The solution also fails to correctly apply the properties of modular inverses and does not properly evaluate the sum. </reason>
(3) <skill> modular_arithmetic, number_theory, understanding_of_fractions </skill>

### C.3.3 Proposing new questions from skill-based training question distribution

In Stage 3 of STAT-Syn, we use GPT-4o to propose new questions based on existing questions in the skill-based training question distribution. Below we show an example prompt.

---

**Example prompt for proposing new questions using GPT-4o**

[TASK]
You'll be given three math questions (e.g., [QUESTION 1]), with their solutions for reference. Your task is to output a new, novel math question that emphasizes the use of [SKILL].
Important Note: the new question should not be very similar to any of the given questions (e.g., naive adaptions like altering variable names or values from a given question is strictly prohibitted). Meanwhile, the new question should have similar difficulty with the given questions.
Output format:
(1) <reason> reason here how the given questions relates to the [SKILL] </reason>
(2) <draft> reason here how to design a new, novel question while emphasizing the [SKILL] </draft>
(3) <question> your newly constructed math question </question>

[QUESTION 1]
{train_set_question1}

[QUESTION 2]
{train_set_question2}

[QUESTION 3]
{train_set_question3}

[SKILL]
{missing_skill}

---

# D ADDITIONAL RESULTS

## D.1 EVALUATION RESULTS ON LLAMA-3.2-1B-INSTRUCT

Table 7 shows the evaluation results on Llama-3.2-1B-Instruct. Similar to Table 3, STAT consistently outperforms both heuristic-based and embedding-based data augmentation baselines on in-distribution dataset and most OOD benchmarks. We presented more discussion in Section 3.2 and Section 4.

| Models | MATH | MATH$^D$ | MATH$^2$ | GSM8K | AMC23 | MATH-perturb | | AIME | | Avg. |
|---|---|---|---|---|---|---|---|---|---|---|
| | | | | | | simple | hard | 2024 | 2025 | |
| *Llama-3.2-1B-Instruct + SFT* | | | | | | | | | | |
| Base Model | 26.0 | 15.1 | 9.1 | 40.7 | 11.1 | 17.2 | 6.5 | 20.0 | 10.0 | 17.3 |
| MATH-Train | 27.0 | 14.5 | 10.0 | 42.8 | 8.8 | 19.0 | 6.8 | **26.7** | 10.0 | 18.4 |
| MATH-Augment | 27.8 | 14.2 | 8.1 | 43.4 | 11.1 | 17.9 | 6.8 | **26.7** | 3.3 | 17.7 |
| MATH-Hard | 28.4 | 15.4 | 8.6 | 44.6 | 10.8 | 18.6 | 7.2 | 23.3 | 3.3 | 17.8 |
| Embed-Sel | 27.4 | 15.6 | 8.6 | 44.6 | 8.8 | 18.6 | 6.8 | **26.7** | 3.3 | 17.8 |
| Embed-Syn | 28.4 | 17.2 | 11.0 | 44.3 | 10.0 | 20.1 | **7.9** | 23.3 | 6.7 | 18.8 |
| **STAT-Sel** | 32.4 | 15.6 | 11.0 | 45.0 | **12.0** | 19.4 | **7.9** | **26.7** | 16.7 | **20.7** |
| **STAT-Syn** | **34.5** | **18.3** | **12.4** | **45.6** | 11.0 | **20.8** | 7.5 | 23.3 | 10.0 | 20.4 |
| *+ GRPO* | | | | | | | | | | |
| Base Model | 31.8 | 14.4 | 9.5 | 49.7 | 13.3 | 23.3 | 8.2 | 20.0 | 6.7 | 19.7 |
| MATH-Train | 32.0 | 16.0 | 11.9 | 50.8 | 10.0 | 23.7 | 7.9 | 16.7 | 6.7 | 19.5 |
| MATH-Augment | 31.2 | 15.0 | 9.0 | 49.1 | 13.6 | **24.7** | 7.9 | 23.3 | 13.3 | 20.8 |
| MATH-Hard | 32.2 | 14.8 | 11.0 | 50.6 | 11.6 | 22.9 | 6.5 | 26.7 | 10.0 | 20.7 |
| Embed-Sel | 32.8 | 16.2 | 11.4 | 49.9 | 12.0 | 21.9 | 6.5 | 23.3 | 13.3 | 20.8 |
| Embed-Syn | 32.6 | 15.0 | 10.5 | **51.0** | 13.9 | 21.1 | 6.8 | 26.7 | 3.3 | 20.1 |
| **STAT-Sel** | 34.8 | 16.6 | 13.8 | 50.1 | **14.8** | 23.7 | **9.0** | 30.0 | 16.7 | 23.3 |
| **STAT-Syn** | **35.2** | **21.1** | 13.8 | **51.0** | **14.8** | **24.7** | 7.9 | **33.3** | 16.7 | **24.3** |

Table 7: Improvements on various math benchmarks from applying STAT. Results under '+SFT' show the performance of SFT models trained with each method, while '+GRPO' shows the performance after applying GRPO on top of the corresponding SFT models. Our methods, STAT-Sel and STAT-Syn, achieve an average gain of up to 3.4% over the base model, with strong OOD performances (AMC23 results reported on average@64, AIME on pass@64). Applying GRPO on top of fine-tuning with STAT further enhances these improvements. See Table 3 for results on Llama-3.2-3B-Instruct and Qwen2.5-3B.

## D.2 MISSING-SKILL-PROFILE

Figure 6 shows the snippets of model-specific Missing-Skill-Profile of Llama-3.2-3B-Instruct, Llama-3.2-1B-Instruct, and Qwen2.5-3B, obtained at the end of Stage 2 (see Section 2.2). These profile snippets include the Top 10 frequent missing skills of the models. As discussed in Section 4, most of the frequent missing skills in both models are algebra-related, such as solving equations, manipulation, and calculation. In addition, both models also demonstrate noticeable weaknesses in conceptual and reasoning-oriented mathematical skills, including combinatorics, understanding and application of functions, and number theory. Compared to Llama-3.2-3B-Instruct, the missing skill profile of Llama-3.2-1B-Instruct concentrated more towards basic operations (e.g., solving equations), suggesting that smaller models have more pronounced limitations in fundamental computational abilities.

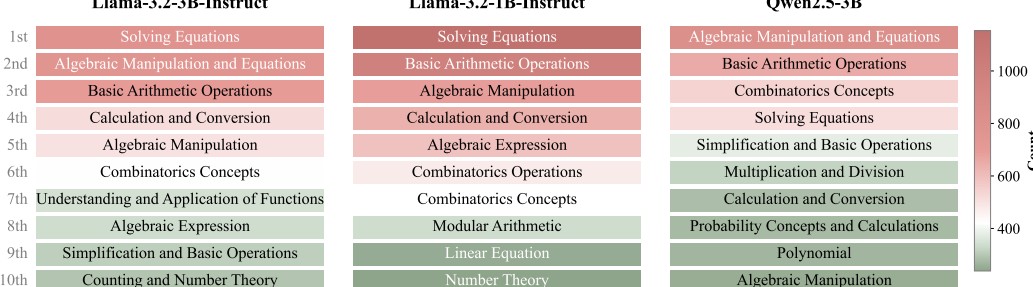

Figure 6: Top 10 missing skills of Llama-3.2-3B-Instruct, Llama-3.2-1B-Instruct, and Qwen2.5-3B. The models struggle most with fundamental mathematical skills such as solving equations and basic arithmetic operations.

### D.3 CASE STUDY: EXAMPLE QUESTIONS WITH DIFFERENT SKILLS

We observe in Section 4 that the baseline training data distribution largely misaligns with model's actual missing skills, with baseline data emphasizing more advanced and conceptual skills (e.g., Prime Number Theory, Polynomial Concepts) while the model lacking more basic operational skills (e.g., Solving Equations). To better visualize this misalignment in data distribution, here we showcase three example QA pairs from the MATH training set, respectively associated with the three skills above.

---

**Question with skill "Solving Equations"**

**Question:** The inverse of $f(x) = \frac{2x-1}{x+5}$ may be written in the form $f^{-1}(x) = \frac{ax+b}{cx+d}$, where $a$, $b$, $c$, and $d$ are real numbers. Find $a/c$.

**Solution:**
If we substitute $f^{-1}(x)$ into our expression for $f$, we get

$$f(f^{-1}(x)) = \frac{2f^{-1}(x) - 1}{f^{-1}(x) + 5}.$$

Since $f^{-1}(f(x)) = x$, we obtain

$$\frac{2f^{-1}(x) - 1}{f^{-1}(x) + 5} = x$$
$$\implies 2f^{-1}(x) - 1 = x(f^{-1}(x) + 5)$$
$$\implies 2f^{-1}(x) - 1 = xf^{-1}(x) + 5x.$$

Moving the terms involving $f^{-1}(x)$ to the left-hand side and the remaining terms to the right-hand side, we get

$$2f^{-1}(x) - xf^{-1}(x) = 5x + 1$$
$$\implies f^{-1}(x)(2 - x) = 5x + 1$$
$$\implies f^{-1}(x) = \frac{5x + 1}{-x + 2}.$$

Now we can see that $(a, b, c, d) = (5, 1, -1, 2)$ for this representation of $f^{-1}(x)$, so

$$\frac{a}{c} = \frac{5}{-1} = \boxed{-5}.$$

---

---

**Question with skill "Prime Number Theory"**

**Question:** The positive integers $A$, $B$, $A - B$, and $A + B$ are all prime numbers. The sum of these four primes is
- A. even
- B. divisible by 3
- C. divisible by 5
- D. divisible by 7
- E. prime

Express your answer using a letter, as A, B, C, D, or E.

**Solution:** The numbers $A - B$ and $A + B$ are both odd or both even. However, they are also both prime, so they must both be odd. Therefore, one of $A$ and $B$ is odd and the other even. Because $A$ is a prime between $A - B$ and $A + B$, $A$ must be the odd prime. Therefore, $B = 2$, the only even prime. So $A - 2$, $A$, and $A + 2$ are consecutive odd primes and thus must be 3, 5, and 7. The sum of the four primes 2, 3, 5, and 7 is the prime number 17, so the correct answer is $\boxed{\text{(E),}}$ prime.

---

**Question with skill "Polynomial Concepts"**

**Question:** The polynomial $P(x) = 2x^3 + ax^2 + bx + c$ has the property that the mean of its zeros, the product of its zeros, and the sum of the coefficients are all equal. The $y$-intercept of the graph of $y = P(x)$ is 8. What is $b$?

**Solution:** The $y$-intercept of the graph is the point at which $x = 0$. At that point, $P(x) = c$, which we are told is equal to 8. Thus, $c = 8$. The product of the roots of the given polynomial is $-\frac{c}{2} = -4$. The problem states that the mean of the zeros must also equal $-4$, so the sum of the three zeros (this is a cubic equation) is equal to $3 \cdot -4 = -12$. The sum of the zeros is also equal to $-\frac{a}{2}$, so $a = 24$. Finally, we are given that the sum of the coefficients, or $2 + a + b + c$, is also equal to $-4$. Plugging in our known values of $a$ and $c$, we have $2 + 24 + b + 8 = -4$. Solving for $b$, we get $b = \boxed{-38}$.

---

## D.4 Effectiveness of STAT on each subject

To evaluate whether STAT enhances general subject-level competence, we measure model accuracy across the 7 subject categories in MATH. These subjects are: prealgebra, algebra, intermediate algebra, geometry, precalculus, number theory, and counting & probability. As shown in Figure 7, both STAT-Sel and STAT-Syn consistently outperform the base model and data augmentation baselines across nearly all subjects. Notably, STAT-Sel achieves the strongest improvements in precalculus and number theory, while STAT-Syn excels in intermediate algebra, prealgebra, algebra, geometry and counting & probability. It is worth noting that STAT brought most improvements on the 3 algebra-related subjects. This aligns with our observation in Section 4 that Llama-3.2-1B-Instruct shows its most pronounced weaknesses in algebra, and confirms that our approaches effectively target the skills the model fundamentally lacks.

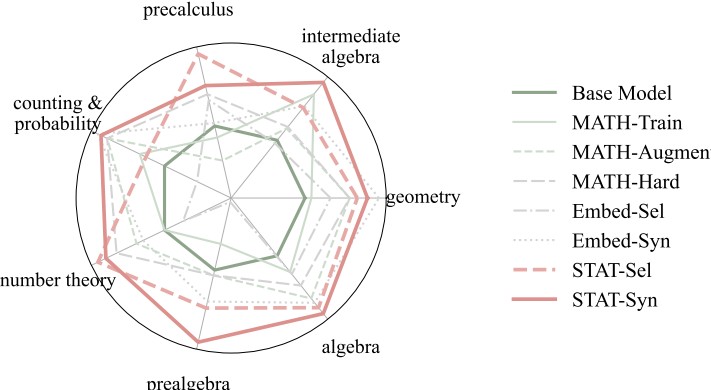

Figure 7: Fine-tuned model performances on MATH subjects, across different training methods. For better visualization, accuracies are normalized per skill axis, with the base model drawn as a uniform circle and the highest-performing method on each skill placed at the outer edge. STAT-Syn and STAT-Sel are most effective in enhancing model performance across nearly all the subjects.

# E    Ablation & Analysis

## E.1 Ablations on the reward filtering method in Stage 1

Recall that in Stage 1 of the STAT pipeline, we use an off-the-shelf process reward model (RLHFlow/Llama3.1-8B-PRM-Mistral-Data) to score small language models' responses, in order to filter out a set of *difficult* questions for each model. Here, we conduct various ablation studies on the reward filtering process.

**Effect of threshold values on the reward model prediction.**    We investigated the effect of $\tau_1$ and $\tau_2$ (defined in Section 2.1) on the classification performance of *difficult* questions. Specifically, we measure whether our classification of questions as *difficult* also corresponds to the correctness of responses assessed using ground-truth labels. In Table 8, we report four metrics (accuracy / precision / recall / F1) evaluating the prediction accuracy resulting from different filtering thresholds. Note that $\tau_1 = 0$ or $\tau_2 = 0$ means completely removing the constraints of $\tau_1$ or $\tau_2$. Across all evaluated combinations of threshold values, our choice of the threshold values ($\tau_1 = 0.85$, $\tau_2 = 0.7$) gives a good combination of prediction scores. To further visualize this effect, we conduct STAT on top of all combinations of thresholds, and report the final accuracy in Table 9. Our choice of threshold values yields the highest final accuracy among all the combinations.

**Out-of-distribution (OOD) prediction performance of reward model.**    Although we primarily evaluated STAT on MATH and GSM8K, our method can potentially be extended to other math datasets. While the reward model we used in Stage 1 was only trained on the MATH and GSM8K distribution, we show that it is capable of scoring responses for various OOD math datasets. Table 10 reports the reward model's performance on classifying correct/incorrect responses from Qwen2.5-3B

| $\tau_1 \backslash \tau_2$ | $\tau_2 = 0$ | $\tau_2 = 0.6$ | $\tau_2 = 0.7$ | $\tau_2 = 0.8$ |
|---|---|---|---|---|
| $\tau_1 = 0$ | 53 / 0 / 0 / 0 | 80 / 78 / 79 / 79 | 80 / 74 / 88 / 79 | 75 / 66 / 95 / 78 |
| $\tau_1 = 0.8$ | 80 / 79 / 78 / 79 | 80 / 76 / 85 / 80 | 79 / 72 / 90 / 80 | 75 / 66 / 96 / 78 |
| $\tau_1 = 0.85$ | 79 / 74 / 88 / 80 | 79 / 72 / 90 / 80 | **78 / 70 / 92 / 80** | 74 / 65 / 96 / 78 |
| $\tau_1 = 0.9$ | 73 / 64 / 95 / 77 | 73 / 64 / 95 / 77 | 72 / 64 / 96 / 77 | 70 / 62 / 97 / 75 |

Table 8: Reward model performance (accuracy / precision / recall / F1) on classifying correct/incorrect responses from Qwen2.5-1.5B-Instruct on MATH, accross different thresholds. $\tau_1 = 0$ or $\tau_2 = 0$ means completely removing $\tau_1$ or $\tau_2$. Our choice of threshold values ($\tau_1 = 0.85, \tau_2 = 0.7$) gives a good combination of prediction scores.

| $\tau_1 \backslash \tau_2$ | $\tau_2 = 0$ | $\tau_2 = 0.6$ | $\tau_2 = 0.7$ | $\tau_2 = 0.8$ |
|---|---|---|---|---|
| $\tau_1 = 0$ | 52.8 | 55.7 | 55.9 | 55.7 |
| $\tau_1 = 0.8$ | 55.1 | 56.3 | 56.2 | 55.6 |
| $\tau_1 = 0.85$ | 55.3 | 56.4 | **56.4** | 55.6 |
| $\tau_1 = 0.9$ | 55.7 | 55.7 | 55.6 | 55.2 |

Table 9: Final STAT performance of Qwen2.5-1.5B-Instruct on MATH, with different thresholds. Our choice of threshold values ($\tau_1 = 0.85, \tau_2 = 0.7$) leads to the highest accuracy.

on four popular math benchmarks: AMC23, AIME24, AIME25, and MATH[2]. The reward model achieves comparably high performance on scoring model responses on these OOD, significantly more difficult benchmarks, indicating that the model is highly generalizable. This implies the potential to extend our method to new datasets without the need to train a specialized reward model for each one.

| Metric | AMC23 | AIME24 | AIME25 | MATH[2] |
|---|---|---|---|---|
| Accuracy | 92.5 | 86.7 | 86.7 | 84.8 |
| Precision | 90.9 | 92.6 | 86.7 | 95.2 |
| Recall | 95.2 | 92.6 | 100.0 | 88.5 |
| F1 | 93.0 | 92.6 | 92.9 | 91.0 |

Table 10: Reward model prediction metrics across four OOD math benchmarks. Despite not being trained on these benchmarks, the reward model's prediction capability is largely generalizable to them.

**Reward Filtering vs. Simple Heuristics for classifying difficult questions.** Considering the computational overhead of calling a separate PRM, we explored alternative approaches to classifying questions that rely on computation-free simple heuristics. Specifically, we experimented with two heuristic strategies:

- **Consistency heuristic:** We measure the consistency of the model across five sampled generations per question and classify questions with lower consistency as difficult. Specifically, a question is *difficult* if, among 5 sampled generations, the most common response appears < 2 times.
- **Length heuristic:** We use the length of the model's responses as a proxy and classify questions with longer responses as difficult. Specifically, a question is *difficult* if the average model response length on this question is ≥ 800 words.

Table 11 shows that both heuristics yield reasonably accurate predictions. Moreover, applying STAT on top of these heuristic-classified difficult questions can improve the final accuracy by 2%. However, we leave a more thorough investigation into the robustness and generalizability of these strategies in relation to PRM-based classification for future work.

**Process Reward vs. Outcome Reward.** We also compare the prediction accuracy of our process reward model (PRM) with threshold filtering (see Section 2.1) against directly loading the reward model as an outcome reward model (ORM). Our preliminary experiments indicated 0.9 as the optimal

| Classification method | Classification accuracy |
|---|---|
| Consistency Heuristic | 79.8% |
| Length Heuristic | 74.2% |
| Reward Filtering | 78.0% |

Table 11: Performance of consistency heuristic and length heuristic on classifying difficult questions. The classification accuracy of simple heuristics are on par with the reward filtering method.

threshold for the outcome rewards. With $\tau = 0.9$, the prediction metrics of the ORM are: Precision = 0.54 / Recall = 0.90 / F1 = 0.68, whereas the prediction metrics of the PRM with optimal thresholds are Precision = 0.70 / Recall = 0.92 / F1 = 0.80. Therefore, our method using PRM with threshold filtering is superior to directly using ORM.

## E.2 STATISTICS OF DIFFICULT QUESTIONS

In Stage 1 of STAT (see Section 2.1), we identify a set of *difficult* questions for each individual model using a process reward model along with a filtering heuristic. Table 12 reports the proportions of difficult questions classified for different models in each math domain. Compared to Table 3, the proportions of difficult questions closely correspond to the accuracy numbers of each model, even though we did not access the ground truth in the whole pipeline. Notably, our classification method captures not only questions that the model gets wrong, but also questions that the model passes with a flawed solution process.

| Model | Geometry | Precalculus | Algebra | Prealgebra | Intermediate Algebra |
|---|---|---|---|---|---|
| Qwen2.5-3B | 61.8 | 70.1 | 29.7 | 33.2 | 75.9 |
| Llama-3.2-1B-Instruct | 93.5 | 92.0 | 91.4 | 89.7 | 99.0 |
| Llama-3.2-3B-Instruct | 68.2 | 82.7 | 45.5 | 48.9 | 85.7 |

| Model | Count.&Prob. | Number Theory | MATH Avg. |
|---|---|---|---|
| Qwen2.5-3B | 62.2 | 56.1 | 52.1 |
| Llama-3.2-1B-Instruct | 97.9 | 95.2 | 94.0 |
| Llama-3.2-3B-Instruct | 65.2 | 62.3 | 62.3 |

Table 12: Proportions of difficult questions (%) classified by STAT for each model. Although our method did not access the ground truth, the proportion of classified difficult questions still closely mirrors each model's accuracy (see Table 3) in each domain.

## E.3 ANALYSIS OF THE TEACHER MODEL

**Teacher model need not be overwhelmingly stronger than student.** One feature of STAT is the demand of a substantially stronger teacher model to supervise the student. In this section, we evaluate this demand by directly comparing teacher and student performances on math reasoning benchmarks. Due to resource constraints, our evaluation is limited to a representative set of benchmarks, but the results are sufficient to illustrate the key trend: the teacher is not strictly dominant, and the student can approach or even match the teacher's performance within a manageable gap.

As shown in Table 13, although teacher models obtain higher absolute scores, they are not overwhelmingly stronger than the students. In particular, the gap between GPT-4o-mini and Qwen2.5-3B is only around 10 points across GSM8K and MATH, a margin that is significant but manageable. This suggests that STAT does not strictly rely on a much stronger teacher to succeed. Instead, even when teacher and student are relatively close in ability, the student can still benefit and recover most of the teacher's performance. This opens up the possibility of self-improvement, where a model iteratively teaches and refines itself without requiring access to an external teacher that is substantially stronger.

**Agreement across different teacher models.** Since our approach relies on a frontier LLM as teacher, a natural concern is potential bias in the missing-skill labeling process. In light of this, we present a preliminary investigation into the level of agreement among different LLMs in missing skill labeling, using an LLM-as-a-judge approach. We first evaluate GPT-4o-mini's ability to self-verify

| Benchmark | Teacher | | Student | | |
|---|---|---|---|---|---|
| | GPT-4o | GPT-4o-mini | Qwen2.5-3B | Llama-3.2-3B-Instruct | Llama-3.2-1B-Instruct |
| GSM8K | 97.0 | 94.0 | 80.9 | 73.0 | 40.7 |
| MATH | 73.0 | 69.1 | 55.8 | 44.0 | 26.0 |
| MATH-perturb-simple | 62.0 | N/A | 43.7 | 33.7 | 17.2 |
| MATH-perturb-hard | 39.4 | N/A | 24.0 | 12.2 | 6.5 |

Table 13: **Math reasoning accuracy (%).** Comparison between teacher models (GPT-4o, GPT-4o-mini) and student models (Qwen2.5-3B, Llama-3.2-3B-Instruct, Llama-3.2-1B-Instruct) on GSM8K, MATH, MATH-perturb-simple, and MATH-perturb-hard.

the correctness of its own predicted missing skills and find that it judges its predictions to be correct 70% of the time. To further assess the reliability of these predictions, we compute the agreement between GPT-4o-mini and Claude-3.5-Sonnet. The models agree on 43% of the predicted skills, where agreement is defined as the average fraction of overlapping skills relative to the total number of skills predicted by GPT-4o-mini. Given the fine-grained nature of our skill list, we consider this level of agreement significant.

