# OpenReview forum: "STAT: Skill-Targeted Adaptive Training"
_ICLR.cc/2026/Conference — ICLR 2026 Poster_

### Official Review · Reviewer_yjVy · 2025-10-31

**Soundness:** 2
**Presentation:** 3
**Contribution:** 2
**Rating:** 4
**Confidence:** 3

**Summary:**

This paper introduces STAT (Skill-Targeted Adaptive Training), a novel fine-tuning framework for language models that leverages the metacognitive abilities of large teacher models to identify and target missing skills in student models. STAT improves generalization and performance on mathematical reasoning benchmarks by either reweighting training data (STAT-Sel) or generating new skill-targeted questions (STAT-Syn). Across multiple benchmarks, STAT consistently outperforms standard SFT and other adaptive data selection methods, and complements reinforcement learning approaches like GRPO.

**Strengths:**

1. The work leverages frontier LLMs' introspective abilities to identify specific skills lacking in student models. This goes beyond error analysis and introduces a more human-like teaching dynamic.
2. STAT methods (both Sel and Syn) demonstrate consistent and significant performance gains over SFT and baselines across in-distribution (MATH) and out-of-distribution tasks (AIME, AMC, MATH-perturb), with gains up to 7.5% on MATH and 5.8% on OOD datasets.

**Weaknesses:**

1. The entire STAT pipeline depends on access to a strong teacher model with metacognitive abilities. This could limit replicability or scalability in low-resource settings or with proprietary LLMs.
2. The Missing-Skill-Profile is built using validation set responses, which might introduce bias if the validation set is not representative of future deployment scenarios.
3. The STAT-Syn variant, while effective, incurs significant computational and inference cost due to on-the-fly synthetic question generation and filtering, which may not be inefficient in large-scale deployments.

**Questions:**

1. How robust is the STAT framework to inconsistencies or ambiguities in the skill taxonomy generated by the teacher model? Would the approach degrade if the teacher’s skill decomposition is noisy or if the skill set is too coarse-grained?
2. Given that STAT relies on a strong LLM teacher to guide training and even generate new examples, how do you mitigate the risk that the teacher’s inductive biases dominate the student’s learning?
3. How do you balance granularity vs. utility when generating the Missing-Skill-Profile? For instance, does distinguishing between "algebraic manipulation" and "linear equation solving" actually improve downstream performance, or introduce noise?

---

> ### Author Response · Authors · 2025-11-22
>
> Thank you for the detailed and helpful feedback! Please find our responses to your questions below.
> ### **“The entire STAT pipeline depends on access to a strong teacher model with metacognitive abilities.”**
>
> 1. In our framework, the teacher model doesn’t need to give the ground truth answers directly to the student. Instead, we only utilize the metacognitive ability of the LLM to gain very high-level feedback on model responses, which **doesn’t rely on an absolutely stronger teacher**.
>    • Below is an example of the extracted LLM output:
>      > algebraic_skills, understanding_circle_properties_and_algebraic_manipulation, coordinate_geometry_and_transformation_skills
>    • Please see Appendix E.3 for detailed discussion of the teacher model.
>
> 2. **We believe that self-improvement (using the same teacher and student model) is possible using our method.** We conduct a small-scale experiment using Llama-3.1-8B-Instruct (referred as Llama-8B below; we use a 8B model because it has better metacognitive abilities than 3B/1B): (1) After identifying difficult questions for Llama-8B, we prompt the model itself to label missing skills for each of its responses; (2) We train Llama-8B using STAT-Sel, where we reweigh the MATH dataset to emphasize frequent missing skills labeled by the model itself.
>    Our key observations are:
>
>    i. **Significant annotation agreement between Llama-3.1-8B-Instruct and GPT-4o.** We obtained the judgement (e.g., “the model response is incorrect”) and missing skill labels from both models, and found that the prediction of Llama-8B has significant agreement with GPT-4o judgement, even though the model size is substantially smaller. This indicates that a smaller model can also output high-quality metacognitive feedback when properly prompted.
>
>    |                    | Accuracy | Precision | Recall | F1   |
>    |--------------------|----------|-----------|--------|------|
>    | Judgement          | 78.5     | 75.0      | 41.2   | 40.1 |
>    | Missing-skill labeling | 51.2 | 44.0      | 51.0   | 47.3 |
>
>    ii. **Self-improvement by STAT-Sel.** We tested the possibility of self-improvement by training Llama-8B using its self-selected STAT-Sel data. As shown below, applying this self-improvement loop results in a performance gain of 6% on MATH and 2% on average, compared to naive fine-tuning. We believe that the gain will be more significant if we apply this self-improvement loop on larger models.
>
>    | Llama-3.1-8B-Instruct | MATH | MATH^D | MATH^2 | gsm8k | AMC23 | MATH-perturb-simple | MATH-perturb-hard | AIME24 | AIME25 | Avg. |
>    |-----------------------|------|--------|--------|-------|-------|----------------------|--------------------|--------|--------|------|
>    | Base model            | 45.7 | 33.1   | 23.8   | 84.1  | 22.8  | 34.4                 | 12.2               | 33.3   | 16.7   | 34.0 |
>    | MATH-train            | 48.0 | 29.8   | 25.2   | 84.6  | 25.0  | 37.6                 | 14.3               | 40.0   | 23.3   | 36.4 |
>    | **STAT-Sel (with self-predicted missing skills)** | **54.0** | **34.0** | **29.0** | **86.2** | **26.6** | **39.8** | **14.3** | **40.0** | **23.3** | **38.6** |
>
> Because self-improvement also results in significant improvements, we can expect a similar training pipeline to also help frontier language models.
>
> ### **“Given that STAT relies on a strong LLM teacher to guide training and even generate new examples, how do you mitigate the risk that the teacher’s inductive biases dominate the student’s learning?”**
> Our goal is to improve the small model’s performance, and leveraging a larger teacher is currently the most effective strategy. Importantly, **STAT-Sel uses an extremely lightweight form of supervision: the teacher provides only skill names, not logits or full rewritten solutions**, as shown in the example output above. This makes it far less intrusive than standard knowledge-distillation methods, where the teacher’s generations or probability distributions can heavily imprint their inductive biases onto the student. By restricting the teacher’s role to identifying relevant skills, STAT-Sel minimizes the risk of the teacher dominating the student’s learning dynamics. Moreover, the self-improvement experiment suggests that the teacher can be replaced with the student themselves, further reducing concerns about external inductive biases influencing the student model’s learning.
>
> ### **“STAT might introduce bias if the validation set is not representative of future deployment scenarios"**
> In our experiments, we show that the improvements on MATH validation also translate to out-of-distribution tasks. Furthermore, continually training the model using missing-skill-profile on the validation set of a harder task (Section 3.3) can lead to further gains. As improvements constructed from the validation set can generalize well, the distribution of the validation set need not perfectly match the eventual deployment distribution.

---

> > ### Author Response · Authors · 2025-11-22
> >
> > ### **"How robust is the STAT framework to inconsistencies or ambiguities in the skill taxonomy generated by the teacher model? How do you balance granularity vs. utility when generating the Missing-Skill-Profile?"**
> > In STAT, our aim was to show that the model’s performance can be substantially improved using the skill ontologies provided by the GPT model in Didolkar et al. [2023]. Because this skill ontology has already been shown to be successful across multiple works [1,2], we built our algorithm around these skill ontologies.
> > Below, we conducted an extra ablation study on the granularity and composition of skills, attempting to reconstruct two other versions of the Skill-Map and re-run the STAT process:
> > 1. **Coarse Skill-Map (20 skills):**
> > We prompted the teacher model to re-cluster the existing 128 skills to derive 20 coarse-grained skills, as follows:
> > |  |  |  |
> > | --- | --- | --- |
> > | Core Arithmetic | Fractions, Decimals, Ratios & Proportions | Exponents, Roots & Logarithms |
> > | Number Theory | Algebraic Expressions & Manipulation | Equations & Inequalities in One Variable |
> > | Systems of Equations & Linear Models | Polynomials & Rational Expressions | Functions, Graphs & Transformations |
> > | Sequences & Series | Complex Numbers | Combinatorics |
> > | Probability | Plane Geometry | Coordinate & Analytic Geometry |
> > | Three-Dimensional Geometry | Trigonometry | Linear Algebra |
> > | Geometric Vector Operations | Introductory Calculus |  |
> >
> > We constructed the coarse Skill-Map on this set of 20 skills and performed **missing skill labeling**. We then trained Llama-3.2-3B-Instruct with STAT-Sel. Our key observations are:
> > - **Reduced but meaningful improvement:** while the original STAT-Sel brings MATH performance from 44.0 to 51.5, coarse-grained STAT-Sel achieves **48.2**.
> > - **Shorter, less targeted missing-skill profiles:** Under the coarse taxonomy, the missing-skill lists from the teacher become substantially shorter, often consisting of one or two skills. This reduces the granularity with which STAT can identify distinct error types.
> > 2. **Fine-grained Skill-Map with skill composition (500 skills):** We prompted the teacher with the current list of 128 skills and asked it to propose all possible compositions of skills. For example, “solving equations” and “geometry” can derive “solving equations derived from geometric diagrams”. We end up with approximately 500 fine-grained skills. However, when attempting to construct the fine-grained Skill-Map, we found that the skill labels were **highly long-tailed**: many skills appeared in 1~2 questions. This made it very difficult to reliably identify missing skills and reweigh train-set data. Therefore, we cannot test our framework on the fine-grained Skill-Map.
> >
> > Thus, our original skill taxonomy (from Didolkar et al. [2023]) strikes a balance in granularity, yielding the strongest gains. Meanwhile, even a much coarser 20-skill map still provides meaningful improvement (from 44.0 to 48.2), which indicates the robustness of our method.
> >
> > ### **“Inefficiency of STAT-Syn for deployment in large-scale deployments”**
> > We proposed 2 variants of STAT: STAT-Sel and STAT-Syn. STAT-Sel is an efficient variant where the teacher simply needs to provide the skills it judges the model to be weak at. **We show that STAT-Sel is already effective in improving the model.**
> >
> > On the other hand, we proposed STAT-syn as an alternative to utilize the teacher to create new synthetic data for training the student. Our primary motivation was to show that STAT can seamlessly be introduced into synthetic data generation frameworks, and can be extremely effective in targeting the key weaknesses of the student. We believe that this direction will become increasingly prominent over time, as we build stronger AI models and require more well-targeted synthetic data in the future.
> >
> > ---
> > [1] Metacognitive Capabilities of LLMs: An Exploration in Mathematical Problem Solving. Didolkar et al. NeurIPS 2024.
> >
> > [2] AdaptMI: Adaptive Skill-based In-context Math Instruction for Small Language Models. He et al. COLM 2025.

---

### Official Review · Reviewer_6kx2 · 2025-11-01

**Soundness:** 3
**Presentation:** 3
**Contribution:** 3
**Rating:** 6
**Confidence:** 4

**Summary:**

The paper proposes STAT, a teacher-student fine-tuning method that uses a stronger LLM to diagnose which skills a smaller model lacks on a math validation set, then adapts training data to target those deficits. Two instantiations are presented: STAT-Sel (re-weights existing examples) and STAT-Syn (synthesizes new questions). On MATH and several out-of-domain (OOD) benchmarks, STAT yields up to 7.5 % absolute gain over vanilla SFT and 4.6 % average OOD improvement. Ablations show that (i) models saturate because they still fail on basic algebraic skills, (ii) embedding-based data selection does not target those skills, and (iii) STAT is complementary to subsequent RL (GRPO).

**Strengths:**

Originality
- First work to operationalize LLM metacognition for training-set adaptation rather than mere evaluation or prompt refinement.
- Skill-map + Missing-Skill-Profile + targeted data construction is a clean pipeline motivated from how humans actually learn.

Empirical Quality
- Experiments on Llama and Qwen families (1B–3B sizes) with MATH as the primary dataset show substantive in-distribution and OOD improvements over naive SFT and embedding-based baselines.
- Careful ablations vs. strong baselines: MATH-Augment, MATH-Hard, Embed-Sel/Syn, and GRPO.
- Hyper-parameters openly tuned per method (sec. 3.1 & C.1).

Clarity
- Algorithm box (Alg. 1), reproducible prompts (Appendix C.3), and released data and code pledge (sec. 7) at camera ready version facilitate replication.

Significance
- Provides a practical recipe to push small models beyond SFT saturation without extra human labels.
- Demonstrates that basic skill gaps (solving linear equations, arithmetic manipulation) remain even after extensive training. I believe these are valuable insight for the community and pushes algorithmic progress towards better data/curriculum design.

**Weaknesses:**

- Dependence on a Powerful Teacher Model: The entire STAT framework is predicated on the availability of a significantly more capable "teacher" LLM (GPT-4o-mini is used) that can accurately diagnose skill deficiencies. How would one go about using this method for improving the skills of frontier models? Is self-improvement a possibility here by using either the same model (prompted in a different way) or by using specialized but smaller models  as teacher?

- Scope limited to MATH-style reasoning (minor): results are strong for math, but applicability to language or safety tasks is only discussed as future work. If the authors could discuss a bit about the generality of the method and possible way to expand it beyond the Math scope, that would be helpful.

- No comparison with gradient-based data selection methods (e.g., RHO-1, Lin et al. 2024) or difficulty-based rejection tuning (DART-math, Tong et al. 2024).

**Questions:**

Based on the comments above, a few questions for the authors:

- Can the skill-map pipeline be automated for non-math domains?
- How many independent training runs (seeds) were performed? If only one, can you provide variance estimates?
- The method's success seems to rely on a predefined list of skills (Section 2, lines 141-145). How sensitive is STAT to the quality and granularity of this skill taxonomy? For instance, would a coarser or finer-grained skill list significantly alter the results?
- How important is using a powerful teacher? Have you considered alternatives which might be better for scaling and help improve fronteir models?

---

> ### Author Response · Authors · 2025-11-22
>
> We would like to thank the reviewer for the detailed feedback on our study. Please find our responses to your questions below.
> ### **“Comparisons with gradient-based data selection methods or difficulty-based rejection tuning”**
> Thank you for pointing us to more baselines. We ran additional experiments for the following baselines:
> 1. **DART-math-uniform:** for each synthesized query, sample until a fixed number of correct responses have been collected. For fair comparison, we randomly sample 9.5k samples from the “hkust-nlp/dart-math-uniform” dataset.
> 2. **DART-math-hard:** assign more correct responses to harder queries. For fair comparison, we randomly sample 9.5k samples from the “hkust-nlp/dart-math-hard” dataset.
> 3. **Gradient-based method:** The cited work is not directly related to gradient-based data selection. We perform a gradient-based experiment mirroring the setup used for Embed-sel: we compute the similarity between difficult validation question–answer pairs and training question–answer pairs, but using model gradients instead of embeddings. It is worth noting that the validation-set answers were generated using GPT-4o, which differs from all of our training strategies and baselines, none of which assume access to ground-truth validation answers. This approach follows the gradient-influence formulation introduced by Pruthi et al. (2020).
> We finetuned Llama-3.2-3B-Instruct using the above datasets with the same training settings described in Section 3.1 in our paper. As shown below, both variants of our method surpass the extra baselines.
> | Llama-3.2-3B-Instruct | MATH | MATH^D | MATH^2 | gsm8k | AMC23 | MATH-perturb-simple | MATH-perturb-hard | AIME24 | AIME25 | Avg. |
> |----------------------|------|--------|--------|-------|-------|----------------------|--------------------|--------|--------|------|
> | Base model         | 44.0 | 18.2 | 21.9 | 73.0 | 21.7 | 33.7 | 12.2 | 33.3 | 16.7 | 30.5 |
> | DART-math-uniform  | 44.0 | 18.2 | 21.9 | 73.0 | 21.7 | 33.7 | 12.2 | 33.3 | 16.7 | 30.5 |
> | DART-math-hard     | 46.2 | 26.3 | 20.4 | 76.3 | 21.8 | 36.1 | 13.3 | 36.7 | 16.7 | 32.6 |
> | Gradient-based     | 48.7 | 27.4 | 19.2 | 78.2 | 22.7 | 37.0 | 13.3 | 26.7 | 23.3 | 32.9 |
> | STAT-Sel           | **51.5** | 26.6 | 25.7 | **80.2** | **24.7** | **39.8** | 13.3 | **43.3** | 23.3 | 36.5 |
> | STAT-Syn           | 50.2 | **31.7** | **26.2** | 79.2 | 23.9 | 39.1 | **14.7** | 40.0 | **30.0** | **37.2** |
> ### **“Would a coarser or finer-grained skill list significantly alter the results?”**
> Yes, but not significantly. Below, we conducted an extra ablation study on the granularity and composition of skills, attempting to reconstruct two other versions of the Skill-Map and re-run the STAT process:
> 1. **Coarse Skill-Map (20 skills):**
> We prompted the teacher model to re-cluster the existing 128 skills to derive 20 coarse-grained skills, as follows:
> |  |  |  |
> | --- | --- | --- |
> | Core Arithmetic | Fractions, Decimals, Ratios & Proportions | Exponents, Roots & Logarithms |
> | Number Theory | Algebraic Expressions & Manipulation | Equations & Inequalities in One Variable |
> | Systems of Equations & Linear Models | Polynomials & Rational Expressions | Functions, Graphs & Transformations |
> | Sequences & Series | Complex Numbers | Combinatorics |
> | Probability | Plane Geometry | Coordinate & Analytic Geometry |
> | Three-Dimensional Geometry | Trigonometry | Linear Algebra |
> | Geometric Vector Operations | Introductory Calculus |  |
>
> We constructed the coarse Skill-Map on this set of 20 skills and performed **missing skill labeling**. We then trained Llama-3.2-3B-Instruct with STAT-Sel. Our key observations are:
> - **Reduced but meaningful improvement:** while the original STAT-Sel brings MATH performance from 44.0 to 51.5, coarse-grained STAT-Sel achieves **48.2**.
> - **Shorter, less targeted missing-skill profiles:** Under the coarse taxonomy, the missing-skill lists from the teacher become substantially shorter, often consisting of one or two skills. This reduces the granularity with which STAT can identify distinct error types.
> 2. **Fine-grained Skill-Map with skill composition (500 skills):** We prompted the teacher with the current list of 128 skills and asked it to propose all possible compositions of skills. For example, “solving equations” and “geometry” can derive “solving equations derived from geometric diagrams”. We end up with approximately 500 fine-grained skills. However, when attempting to construct the fine-grained Skill-Map, we found that the skill labels were **highly long-tailed**: many skills appeared in 1~2 questions. This made it very difficult to reliably identify missing skills and reweigh train-set data. Therefore, we cannot test our framework on the fine-grained Skill-Map.
>
> Thus, our original skill taxonomy with balanced granularity yields the strongest gains, while even a much coarser 20-skill map still provides meaningful improvement.

---

> ### Author Response · Authors · 2025-11-22
>
> ### **“Dependence on a Powerful Teacher Model”**
>
> 1. In our framework, the teacher model doesn’t need to give the ground truth answers directly to the student. Instead, we only utilize the metacognitive ability of the LLM to gain very high-level feedback on model responses, which doesn’t rely on an absolutely stronger teacher.
>
>    • Below is an example of the extracted LLM output:
>      > algebraic_skills, understanding_circle_properties_and_algebraic_manipulation, coordinate_geometry_and_transformation_skills
>
>    • Please see Appendix E.3 for detailed discussion of the teacher model.
>
> 2. We believe that self-improvement (using the same teacher and student model) is possible using our method. We conduct a small-scale experiment using Llama-3.1-8B-Instruct (referred as Llama-8B below; we use a 8B model because it has better metacognitive abilities than 3B/1B): (1) After identifying difficult questions for Llama-8B, we prompt the model itself to label missing skills for each of its responses; (2) We train Llama-8B using STAT-Sel, where we reweigh the MATH dataset to emphasize frequent missing skills labeled by the model itself.
>    Our key observations are:
>
>    i. **Significant annotation agreement between Llama-3.1-8B-Instruct and GPT-4o.** We obtained the judgement (e.g., “the model response is incorrect”) and missing skill labels from both models, and found that the prediction of Llama-8B has significant agreement with GPT-4o judgement, even though the model size is substantially smaller. This indicates that a smaller model can also output high-quality metacognitive feedback when properly prompted.
>
>    |                    | Accuracy | Precision | Recall | F1   |
>    |--------------------|----------|-----------|--------|------|
>    | Judgement          | 78.5     | 75.0      | 41.2   | 40.1 |
>    | Missing-skill labeling | 51.2 | 44.0      | 51.0   | 47.3 |
>
>    ii. **Self-improvement by STAT-Sel.** We tested the possibility of self-improvement by training Llama-8B using its self-selected STAT-Sel data. As shown below, applying this self-improvement loop results in a performance gain of 6% on MATH and 2% on average, compared to naive fine-tuning. We believe that the gain will be more significant if we apply this self-improvement loop on larger models.
>
>    | Llama-3.1-8B-Instruct | MATH | MATH^D | MATH^2 | gsm8k | AMC23 | MATH-perturb-simple | MATH-perturb-hard | AIME24 | AIME25 | Avg. |
>    |-----------------------|------|--------|--------|-------|-------|----------------------|--------------------|--------|--------|------|
>    | Base model            | 45.7 | 33.1   | 23.8   | 84.1  | 22.8  | 34.4                 | 12.2               | 33.3   | 16.7   | 34.0 |
>    | MATH-train            | 48.0 | 29.8   | 25.2   | 84.6  | 25.0  | 37.6                 | 14.3               | 40.0   | 23.3   | 36.4 |
>    | **STAT-Sel (with self-predicted missing skills)** | **54.0** | **34.0** | **29.0** | **86.2** | **26.6** | **39.8** | **14.3** | **40.0** | **23.3** | **38.6** |
>
> Because self-improvement also results in significant improvements, we can expect a similar training pipeline to also help frontier language models.
>
> ### **“Scope limited to MATH-style reasoning; Can the skill-map pipeline be automated for non-math domains?”**
>
> To the best of our knowledge, the skill-map pipeline can be automated for instruction following and coding tasks.
>
> 1. **Instruction following:**
>    Instruct-SkillMix (Kaur et al., 2025) shows that skill-based training using new skill compositions can significantly improve the model’s win rate on various instruction-following tasks. They explored a pipeline of labeling skills (using gpt-4) → clustering skills → synthesizing data with new skill compositions, building on top of existing instruction-following datasets (Alpaca-52k and UltraChat).
>    The skill map consists of **337 skills from Alpaca, 128 skills from UltraChat, and 35 alignment + safety skills**.
>
> 2. **Coding:**
>    We conducted an extra experiment to construct a skill-map for coding tasks, building upon existing datasets like LiveCodeBench. After running skill labeling and clustering, we obtained **11 high-level categories and 360 total skills**. Below we show some example skills in four of the categories.
>
> | Algorithm  | C++ |
> |----------|----------|
> | basic_programming_model, data_abstraction, bags_queues_and_stacks, … | basic_syntax_and_structure_of_c++, functions_and_function_overloading_in_c++, recursion_in_c++, … |
>
> | Functional Programming | JavaScript |
> |-------------|-------------------------|
> | pure_functions_and_immutability, first_class_and_higher_order_functions, anonymous_functions_and_closures, … | javascript_tooling_and_build_workflows, browser_storage_techniques_and_persistence, javascript_cross_browser_support_and_compatibility_handling, … |

---

> ### Author Response · Authors · 2025-11-22
>
> ### **“How many independent training runs (seeds) were performed? If only one, can you provide variance estimates?”**
>
> Thank you for pointing this out. We performed one training run in each of our experiments. In response to the reviewer, we provide variance estimates by conducting three independent runs with different seeds (0, 42, 45) for training data selection. The variance estimates are presented below.
>
> |              | Llama-3.2-3B-Instruct | Llama-3.2-1B-Instruct | Qwen2.5-3B |
> |--------------|------------------------|------------------------|------------|
> | **STAT-Sel** | 51.2 ± 0.3              | 32.4 ± 0.1              | 58.4 ± 0.2  |
> | **STAT-Syn** | 50.5 ± 0.4              | 34.4 ± 0.2              | 59.2 ± 0.4  |

---

### Official Review · Reviewer_PEvi · 2025-11-03

**Soundness:** 3
**Presentation:** 3
**Contribution:** 3
**Rating:** 6
**Confidence:** 5

**Summary:**

Supervised fine-tuning can often result in limited gains. This paper proposes an approach, STAT, to transform an SFT dataset (via reweighting or synthetic data generation) based on skills. Given a validation dataset, the current model's failures on it are analyzed to determine which skills are missing from the model. These skills are linked back to the training data, and data most associated with these skills are upweighted or synthetically generated. Empirical results show that STAT-Syn and STAT-Sel significantly outperform other data-centric approaches on several model families and sizes. It can also be used in continual learning, and also can be used with RL downstream.

**Strengths:**

Originality:
- This paper builds on several existing skills-based works, e.g., using their techniques for skill discovery and skill mapping. However, this is not a significant drawback. Moreover, this work focuses on the SFT setting and uses teachers/skills exclusively for alignment, rather than embeddings or gradients.

Quality:
- Experiments show that STAT can significantly improve performance across model sizes and families for math.

Clarity:
- The paper is well-written, and the method is easy to understand.

Significance:
- A common heuristic for curate good training datasets is to simply create or select samples that "look like" the evaluation data. E.g., if you want your model to do well on science, you need to train on science data. However, this heuristic is fairly imprecise and can fail to capture the underlying skills that samples teach the model. Moving towards skill-based data curation is an important direction, and I foresee datasets being annotated with more and more fine-grained skill taxonomies in the future to facilitate greater gains.

**Weaknesses:**

Quality:
- The method operates under a reasonable assumption that the training data distribution should be aligned with the test data distribution in terms of skills (Figure 2). However, this does not take into account the possibility that 1) skills may interact with each other and 2) skills are learned at different rates (as suggested in Chen et al 2023 https://arxiv.org/abs/2307.14430).
- The continual learning extension showcases the flexibility of STAT; however, I am also curious about if optimizing on MATH-perturb-hard negatively impacts performance on the other math benchmarks. More generally, I am interested in seeing if the model forgets previous skills, and if so, whether you could make some simple adjustments to STAT (e.g., doing an exponential moving average for STAT-Sel over time) to address this.
- I am curious about the role of composition in these results. In Figure 5, STAT-Syn determines that the missing skill is solving equations, while Embed-Syn determines that the missing skill is Ellipse Properties. Is it ever the case that the actual missing skill is a union of these skills? For instance, what is needed is not just more data focused on solving equations, but instead data focused on solving equations related to geometry? This may also touch more fundamentally on a question of skill granularity/combinatorics of skills.
- I am curious about an "oracle" experiment that could track whether or not the missing skills are learned by the model. Here's a proposal: instead of using a validation set, take the test set, and use the ground truth labels to determine the difficult samples, which are just the ones that the model gets wrong. If we apply STAT-Sel/STAT-Syn, how many of the questions in the test set will flip and become correct? Does STAT interfere with the test set questions that were already answered correctly?

Significance:
- The empirical evaluation is somewhat narrow - it relies on a single dataset and a specific skill taxonomy. This makes it unclear how well the findings generalize to other domains (e.g., coding, agentic workflows) or to different definitions of “skill.” In particular, the analysis in this paper suggests that skills are deeper than the topics identified via embeddings (e.g., in Figure 5), but it is not completely clear how to construct these skills from first principles on a new domain besides math.

**Questions:**

(Taken from the Weaknesses section above)
1. Have you considered modeling or empirically testing for interactions or uneven learning rates among skills? How might such dynamics affect the assumptions behind STAT?
2. In the continual learning setup, how does optimizing on MATH-perturb-hard impact performance on other math benchmarks? Do you see any sort of forgetting going on and could that be fixed?
3. Do you observe cases where the true missing capability is a composition or union of multiple skills (e.g., solving equations related to geometry)? Could STAT handle such compositional skill structures?
4. To show STAT's improvement at the skill-level, can you consider an oracle-style evaluation where you identify difficult test samples (those currently misclassified) and measure how many become correct after applying STAT?
5. How do you expect the method to generalize to other domains such as coding, reasoning, or agentic workflows? What guidance could you provide for defining “skills” in new domains, given that the math taxonomy here is well structured?

---

> ### Author Response · Authors · 2025-11-22
>
> We would like to thank the reviewer for the careful review and their positive feedback on our study. Please find our responses to your questions below.
> ### **“Skill granularity and composition of skills.”**
>
> In STAT, our aim was to show that the model’s performance can be substantially improved using the skill ontologies provided by the GPT model in Didolkar et al. [2023]. Because this skill ontology has already been shown to be successful across multiple works [1,2], we built our algorithm around these skill ontologies.
>
> We conducted an extra ablation study on the granularity and composition of skills, attempting to reconstruct two other versions of the Skill-Map and re-run the STAT process:
>
> 1. **Coarse Skill-Map (20 skills):**
> We prompted the teacher model to re-cluster the existing 128 skills to derive 20 coarse-grained skills, as follows:
>
> |  |  |  |
> | --- | --- | --- |
> | Core Arithmetic | Fractions, Decimals, Ratios & Proportions | Exponents, Roots & Logarithms |
> | Number Theory | Algebraic Expressions & Manipulation | Equations & Inequalities in One Variable |
> | Systems of Equations & Linear Models | Polynomials & Rational Expressions | Functions, Graphs & Transformations |
> | Sequences & Series | Complex Numbers | Combinatorics |
> | Probability | Plane Geometry | Coordinate & Analytic Geometry |
> | Three-Dimensional Geometry | Trigonometry | Linear Algebra |
> | Geometric Vector Operations | Introductory Calculus |  |
>
> We constructed the coarse Skill-Map on this set of 20 skills and performed **missing skill labeling**. We then trained Llama-3.2-3B-Instruct with STAT-Sel. Our key observations are:
>
> - **Reduced improvement:** while the original STAT-Sel brings MATH performance from 44.0 to 51.5, coarse-grained STAT-Sel only achieves **48.2**.
>
> - **Shorter, less targeted missing-skill profiles:** Under the coarse taxonomy, the missing-skill lists from the teacher become substantially shorter, often consisting of one or two skills, rather than the more diagnostic fine-grained skills. This reduces the granularity with which STAT can identify distinct error types.
>
> 2. **Fine-grained Skill-Map with skill composition (500 skills):** We prompted the teacher with the current list of 128 skills and asked it to propose all possible compositions of skills. For example, “solving equations” and “geometry” can derive “solving equations derived from geometric diagrams”; “probability” and “combinatorics” can derive “applying permutations in probability”. We end up with approximately 500 fine-grained skills. However, when attempting to construct the fine-grained Skill-Map, we found that the skill labels were **highly long-tailed**: many skills appeared in 1~2 questions. This made it very difficult to reliably identify missing skills and reweigh train-set data. Therefore, we cannot test our framework on the fine-grained Skill-Map.
>
> We therefore conclude that the skill ontology from Didolkar et al. [2023] provides an effective balance of granularity: it incorporates meaningful skill composition while not being long-tailed.
>
> ### **“Consider an ‘oracle’ experiment that tracks whether or not the missing skills are learned by the model.”**
>
> Thanks for suggesting this interesting experiment. We run this experiment with the following process:
> 1. On the whole MATH test set, we label question difficulty using the ground truth, and **label missing skills** for each difficult question.
> 2. We perform STAT-Sel and STAT-Syn accordingly on the labeled sets, and train Llama-3.2-3B-Instruct with the “oracle” data.
> 3. We tested the “oracle” model on **MATH**, and tracked how many difficult questions associated with the Top 10 missing skills flipped and became correct.
>
> Our key results are:
> - **Further increased accuracy on MATH:** Applying STAT under this oracle setting can further boost 2–3% performance.
> | Base model | STAT-Sel | STAT-Syn | STAT-Sel (Oracle) | STAT-Syn (Oracle) |
> |-----------|----------|----------|--------------------|--------------------|
> | 44.0 | 51.5 | 50.2 | **54.0** | **53.8** |
>
> - **Oracle STAT flipped a substantial number of previously incorrect questions into correct ones (False → True)**: More frequent missing skills (e.g., solving equations) exhibit the largest numbers of flipped cases. This suggests that our framework can effectively address the missing skills.
>
> | Top 10 Missing Skills | STAT-Sel  | | STAT-Syn |  |
> |-------|----------------------|----------------------|----------------------|----------------------|
> |       | False→True | True→False | False→True | True→False |
> | Solving Equations | 44 | 12 | 45 | 9 |
> | Basic Arithmetic Operations | 24 | 9 | 20 | 7 |
> | Algebraic Manipulation | 12 | 9 | 13 | 6 |
> | Calculation and Conversion | 7 | 3 | 10 | 2 |
> | Algebraic Expression | 2 | 0 | 2 | 0 |
> | Combinatorics Operations | 3 | 2 | 3 | 1 |
> | Combinatorics Concepts | 1 | 1 | 0 | 0 |
> | Modular Arithmetic | 6 | 2 | 7 | 2 |
> | Linear Equation | 8 | 1 | 8 | 0 |
> | Number Theory | 4 | 7 | 9 | 5 |

---

> ### Author Response · Authors · 2025-11-22
>
> ### **“How do you expect the method to generalize to other domains such as coding, reasoning, or agentic workflows?”**
>
> To the best of our knowledge, the skill-map pipeline can be automated for instruction following and coding tasks.
>
> 1. **Instruction following:**
>    Instruct-SkillMix (Kaur et al., 2025) shows that skill-based training using new skill compositions can significantly improve the model’s win rate on various instruction-following tasks. They explored a pipeline of labeling skills (using gpt-4) → clustering skills → synthesizing data with new skill compositions, building on top of existing instruction-following datasets (Alpaca-52k and UltraChat).
>    The skill map consists of **337 skills from Alpaca, 128 skills from UltraChat, and 35 alignment + safety skills**.
>
> 2. **Coding:**
>    We conducted an extra experiment to construct a skill-map for coding tasks, building upon existing datasets like LiveCodeBench. After running skill labeling and clustering, we obtained **11 high-level categories and 360 total skills**. Below we show some example skills in four of the categories.
>
> | Algorithm  | C++ |
> |----------|----------|
> | basic_programming_model, data_abstraction, bags_queues_and_stacks, … | basic_syntax_and_structure_of_c++, functions_and_function_overloading_in_c++, recursion_in_c++, … |
>
> | Functional Programming | JavaScript |
> |-------------|-------------------------|
> | pure_functions_and_immutability, first_class_and_higher_order_functions, anonymous_functions_and_closures, … | javascript_tooling_and_build_workflows, browser_storage_techniques_and_persistence, javascript_cross_browser_support_and_compatibility_handling, … |
>
> ### **“Does optimizing on MATH-perturb-hard negatively impact performance on the other math benchmarks?”**
>
> We evaluate STAT-ConSel and STAT-ConSyn from section 3.3 on other MATH datasets.
> Recall that these models were trained with skill profiles built on the validation set of MATH-perturb-hard.
>
> | Llama-3.2-3B-Instruct | MATH | MATH^D | MATH^2 | gsm8k | AMC23 | MATH-perturb-simple | MATH-perturb-hard | AIME24 | AIME25 | Avg. |
> |----------------------|------|--------|--------|-------|-------|----------------------|--------------------|--------|--------|------|
> | Base model      | 44.0 | 18.2 | 21.9 | 73.0 | 21.7 | 33.7 | 12.2 | 33.3 | 16.7 | 30.5 |
> | STAT-Sel        | **51.5** | 26.6 | 25.7 | **80.2** | 24.7 | 39.8 | 13.3 | **43.3** | 23.3 | 36.5 |
> | **STAT-ConSel**     | 51.0 | 28.8 | 27.8 | 78.2 | **25.2** | 38.9 | 17.2 | 40.0 | 23.3 | 36.7 |
> | STAT-Syn        | 50.2 | 31.7 | 26.2 | 79.2 | 23.9 | 39.1 | 14.7 | 40.0 | **30.0** | 37.2 |
> | **STAT-ConSyn**     | **51.5** | **32.5** | **28.0** | **80.2** | 24.9 | **40.3** | **17.6** | 40.0 | **30.0** | **38.3** |
>
> As presented in the results, we observe that the effect on the other datasets is negligible for STAT-ConSel.
> For STAT-ConSyn, results on all datasets are slightly better than STAT-Syn, proving that adding more targeted synthetic data that targets missing skills on a different dataset delivers benefits overall.
>
> ### **“Modeling or empirically testing for interactions or uneven learning rates among skills?”**
>
> We agree about the uneven learning rates and interactions among skills. The appearance of ‘missing skills’ reflects this imbalance: uniform fine-tuning leaves slow-learning skills underdeveloped. STAT directly targets these under-learned skills by upweighting them in the training distribution, yielding gains even for heavily trained models. Although we did not explicitly model skill interactions, this is a promising direction. Prior work such as Chen et al. (Skill-it) studies uneven learning rates and interactions among manually defined skills; extending such analyses to our meta-cognitive skill structure would be a next step for future work.
>
> ---
> [1] Metacognitive Capabilities of LLMs: An Exploration in Mathematical Problem Solving. Didolkar et al. NeurIPS 2024.
>
> [2] AdaptMI: Adaptive Skill-based In-context Math Instruction for Small Language Models. He et al. COLM 2025.

---

### Meta-Review · Area_Chair_ACP9 · 2025-12-29

**Summary:**

Summary of major concerns:

1. Limited empirical evidence, since the experiments rely on just one domain (math), and one taxonomy. (Reviewers PEvi and 6kx2)
2. Missing comparisons with other data selection approaches, particularly gradient-based and difficulty-based selection approaches. (Reviewer 6kx2)
3. This method relies on a strong teacher. So it is unclear if it would work for training frontier models. (Reviewers 6kx2 and yjVy)
4. The original experiments keep the skill taxonomy fixed, and do not study the effect of the granularity of the taxonomy, and the composition of skills. (Reviewers PEvi, 6kx2).
5. Unclear how the method impacts forgetting untargeted skills in the same domain. (Reviewer PEvi)

**Reviewer Concerns:**

**Addressed by the rebuttal:** The authors provided results from new experiments comparing the proposed method with DART (a difficulty based selection approach) and selection based on gradient similarity, and the results address (2) above. They also provided results using Llama 3.1 8B instead of a strong teacher, that also addresses (3) well. I believe (4) is also addressed by the new results in the response which show that a coarser taxonomy leads to worse results. Similarly, the results showing performance on other meth benchmarks addresses (5).

**Outstanding concerns:** I believe (1) is an important limitation, and is not fully addressed. The authors' response provides pointers to prior work (Kaur et al., 2025) that used a taxonomy based approach to generate data for new skills in other tasks, and also showed a synthesized taxonomy in a new domain (coding), but none of these convincingly shows the applicability of the proposed method (synthesizing data targeting missing skills of a model) in additional domains.

In addition, a concern that the reviewers had not brought up is that of the significance of the contributions in this paper. I think that is a valid additional concern, given that there is prior work which synthesizes training data targeting specific skills in a domain (Kaur et al., 2025) and that the actual taxonomy used in this work is also from prior work (Didolkar et al., 2024).

**Reviewer Scores:**

I think there is a chance that all three reviewers would have increased their scores given that the responses satisfactorily addresses at least one of each reviewers concerns.

---

### Decision · Program_Chairs · 2026-01-26

Accept (Poster)